



# Identifying and quantifying geogenic organic carbon in soils – the case of graphite

Jeroen H.T. Zethof [1], Martin Leue[2], Cordula Vogel[1], Shane W. Stoner[3], Karsten Kalbitz[1]

[1] Institute of Soil Science and Site Ecology, Technische Universität Dresden, 01737 Tharandt, Germany
[2] Leibniz-Centre for Agricultural Landscape Research (ZALF), Research Area 1 Landscape Functioning, Working Group Hydropedology, Eberswalder Str. 84, D-15374 Müncheberg, Germany
[3] Max-Planck Institute for Biogeochemistry, Hans-Knöll-Straße 10, 07745 Jena, Germany

*Correspondence to*: Jeroen Zethof (jeroen.zethof@tu-dresden.de)

**Abstract**

A widely overlooked source of carbon (C) in the soil environment is organic carbon (OC) of geogenic origin, e.g. graphite, occurring mostly in metamorphic rocks. Appropriate methods are not available to quantity graphite and to differentiate it from other organic and inorganic C sources in soils. This methodological shortcoming also complicates studies on OC in soils formed on graphite-containing bedrock, because of the unknown contribution of a very different soil OC source.

In this study, we examined Fourier-transform infrared (FTIR) spectroscopy, Thermogravimetric analysis (TGA) and the smart combustion methods for their ability of identifying and quantifying graphitic C in soils. For this purpose, several artificial soil samples with graphite, $CaCO_3$ and plant litter as usual C components were created. A graphitic standard was mixed with pure quartz and a natural soil for calibration and validation of the methods over a graphitic C range of 0.1 to 4%. Furthermore, rock and soil material from both a graphite bearing schist and a schist without natural graphite were used for

method validation.

*FTIR*: As specific signal intensities of distinct graphite absorption bands were missing, calibration could only be performed on general effects of graphite contents on the energy transmitted through the samples. The use of samples from different mineral origin yielded significant matrix effects and hampered the prediction of geogenic graphite contents in soils.

*TGA*: Thermogravimetric analysis, based on changes in mass loss due to differences in thermal stabilities, are
suggested as a useful method for graphite identification, although (calcium) carbonate and graphitic C have a similar thermal stability. However, the quantitative estimation of the graphite contents was challenging as dehydroxylation (mass loss) of a wide range of soil minerals occur in a similar temperature range.

*Smart combustion*: The method is based on measuring the release of C during a combustion program, quantified by a non-dispersive infrared detector (NDIR) being part of a commercial elemental analyser, whereby carbonates and graphitic C
could be separated by switching between oxic and anoxic conditions during thermal decomposition. Samples were heated to 400 °C under oxygen rich conditions, after which further heating was done under anoxic conditions till 900 °C. The residual oxidizable carbon (ROC), hypothesized to be graphitic C, was measured by switching back to oxygenic conditions at 900 °C. Test samples showed promising results for quantifying graphitic C in soils. For the purpose of quantifying graphitic C content





in soil samples, smart combustion was the most promising method of those who have been examined in this study. However, caution should be taken with carbonate rich soils as increasing amounts of carbonate resulted in an underestimation of graphitic C content.

## 1 Introduction

In the past decades, global carbon (C) cycling has gained more and more attention. As an important component in this cycle, the soil C reservoir consists of many different forms and types of carbonaceous substances, each with unique turnover times and functions. Probably the most widely studied and dynamic soil C component is fresh organic matter (OM), derived from litter input, decaying organisms and plant exudates (summarized in Blankinship et al., 2018). Another well studied, soil C component is inorganic C in the form of carbonates, which form an important part of the soil C pool, especially under arid

climate conditions (Zamanian et al., 2016; Apesteguie et al., 2018). Black C, defined as a broad set of highly condensed carbonaceous by-products (e.g. soot) and residues (e.g. charcoal) of incomplete fossil fuel and biomass combustion, has obtained an increasing interest during the past decades (Agarwal and Bucheli, 2011). Organic C (OC) of geogenic origin, which gained less attention until now, is formed when organic compounds in sediments undergo coalification or kerogen transformation during diagenesis. Under high pressure and appropriate temperature conditions this process can continue into

the formation of graphitic C (Oohashi et al., 2012; Buseck and Beysacc, 2014). Intruding hydrothermal fluids in the earth's crust forms a second source of graphitic C during rock formation (Rumble, 2014). This pure and stable form of C is highly chemical inert, although impurities from the parent material increases its chemical reactivity (Beyssac and Rumble, 2014). Via tectonic processes graphite bearing rocks can reach the earth surface where they are subjected to physical and chemical weathering. The fate of this geogenic graphite under weathering and soil formation has rarely been studied, possibly due to the

lack of methods for determining and quantifying geogenic graphite beyond the background of soil OC (OC). There are some indications that a substantial part of the geogenic graphitic C is actually lost in the pathway from rock weathering to (marine) sedimentation (Galy et al., 2008; Clark et al., 2017). In a recent study, Hemmingway et al. (2018) estimated 2/3 of the graphitic C to get oxidized during soil formation, strongly facilitated by soil microbial activity.

The necessity of identifying and quantifying geogenic C becomes obvious when considering the widely used [14]C

dating method to measure the mean age of substances and their turnover rates (Trumbore, 2000). As C is depleted in [14]C over 50.000 years following burial, geogenic C will contain no longer [14]C and might dilute the [14]C content of younger C pools (Rumpel and Kögler-Knabner, 2011). Although the dilution effect might be of less importance for the C pool in topsoil, in subsoil this can become more important as the C gets older and geogenic C might have a more dominant share in the total C pool (Rumpel and Kögler-Knabner, 2011). If geogenic C cannot be distinguished from the "normal" soil organic C derived

from fresh OM, age and turnover time of soil OC will be overestimated. For instance, in the case of Hemmingway et al. (2018), the OM in the upper soil layers should have aged for over 20.000 years to explain the derived [14]C signal, which is unlikely under the local tropical conditions. Although exact figures are unknown, more than one fifth of the global lithology has the

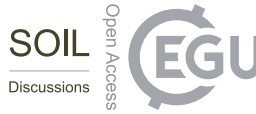

potential of containing graphitic C (Hartmann and Moosdorf, 2012). This illustrates the necessity to distinguish between the different C sources to be able to study their fate and residence time.

Several quantification methods, based on optical, thermal and chemical properties, have been established for identification and quantification of various C sources. Probably the most widely applied method is measuring C released after

dry combustion. However, when different (in)organic C components are present, they might not be differentiated by dry combustion and should therefore be corrected for or even removed. Several pre-treatments, like thermal differentiation (e.g. Apesteguia et al., 2018) or removal by acid fumigation (e.g. Harris et al., 2001) in the case of carbonates, have been established to differentiate between the different carbonaceous substances.

Other methods, such as Fourier-transform infrared (FTIR) spectroscopy in the mid-infrared range (wave lengths 2.5

– 25 μm) is a widely used technique to qualify organic and mineral matter in soils in terms of its functional groups (Smith, 1995; Parikh et al., 2014). Transmission FTIR yields highly resolved spectra with clearly separated absorption peaks, but requires sample dilution. In contrast, diffuse reflectance infrared Fourier transform (DRIFT) can be applied to undiluted soil samples (Reeves, 2003), in particular to determine OC contents of soils (e.g. Reeves et al., 2002; McCarty et al., 2002) via partial least squares regression (PLSR) (e.g. Janik et al., 1998). For employing PLSR, DRIFT spectra are calibrated by OC

contents obtained with standard techniques such as dry combustion (e.g. Vohland et al., 2014).

Mid-infrared spectra from graphite show few absorption bands. Among the bands at wave numbers 2200, 1587, 1362, and 830 cm$^{-1}$ reported by Friedel and Carson (1971), the bands at 1587 and 868 cm$^{-1}$ were attributed to optical lattice vibrational modes of graphite (Chung 2002) while the other two bands have not been found. Tan et al. (2013) reported no prominent peak in FTIR spectra from pure graphite powder. However, in case of oxidized or impure graphite, a number of infrared absorption

bands assigned to C–OH (3400 cm$^{-1}$), C=O (1729 cm$^{-1}$), phenolic C–OH (1220 cm$^{-1}$), C–O (1052 cm$^{-1}$) and aromatic C–H (870 cm$^{-1}$) have been reported (Tan et al., 2013). Depending on the graphite C amount and transformation stage it is not clear until now, if they can be defined in soil samples.

Thermal / thermogravimetric analyses (TGA) have been applied for a long time to study the mineral components of soils and rocks. For instance, the Rock-Eval method has been developed for oil and gas exploration, whereby the hydrocarbon,

CO and $CO_2$ concentrations are measured during a consecutive pyrolysis and oxidation program under constant heating (Behar et al., 2001). More recent thermal analyses have been adopted to study the oxidative behaviour of soil OC, which might serve as a proxy for biogeochemical stability of these substances (Plante et al., 2009). The Rock-Eval method has been successfully applied to characterize the more stable part of OM remaining in the soil after long-term bare fallow (Barré et al., 2016). One of the advantages of TGA is the relatively inexpensive approach with minimal sample preparation needed to distinguish

between different soil C components (Plant et al., 2009; Fernández et al., 2012; Kučerík et al., 2018). Additionally, it is a promising method to differentiate between the thermally instable OM and high stable geogenic C, like coal or graphite. A precondition for quantifying substances with the TGA method is that the thermal properties of the substance of interest are known, i.e. the temperature limits at which the oxidation/decomposition reactions take place. As no universally accepted





temperature limits currently exists, the method still depends on empirically derived temperature boundaries to differentiate and quantify substances (Pallasser et al., 2013; Ussiri et al., 2014).

It has been demonstrated by Fernández et al. (2012) that TGA coupled with differential scanning calorimetry and evolved gas analysis ($CO_2$/$H_2O$), increase the accuracy of quantifying organic substances during thermal analysis. Especially,
the presence of $CaCO_3$ could be detected in smaller quantities as with conventional TGA the decomposition could be masked by the dihydroxylation of (clay) minerals (Fernández et al., 2012). The same can also be expected for oxidation of graphitic C, as it takes place at roughly the same temperature ranges as the (clay) dihydroxylation and $CaCO_3$ decomposition (Hayhurst and Parmar, 1998; Bews et al., 2001).

Recently, a new method has been developed based on experience with TGA measurements, which is defined in the
DIN 19539-standard (DIN Standards Committee Water Practice, 2016). In short, the DIN-standard defines biologically labile OM in solid samples, including soils, to be thermally oxidized at temperatures below 400°C ($TOC_{400}$), while residual oxidizable C (ROC), like lignite or soot, and inorganic C ($TIC_{900}$) are oxidized respectively decomposed between 400 and 900 °C. Combustion elemental analysers, based on this DIN-standard, offer also the possibility to alternate between oxic and anoxic conditions during a measurement. In this method, dubbed "smart combustion", C components are consequently differentiated
on both thermal and oxidizable properties. In theory, graphite, a pure C, will oxidize poorly under anoxic conditions (Hayhurst and Parmar, 1998; Bews et al., 2001), while carbonates do not require oxygen to decompose at these temperatures. Contradictory to the pyrolyzing step of OM by the Rock-Eval method (Behar et al., 2001), OM is immediately oxidized in the first heating phase with the smart combustion method. Therefore, it is less likely that by-products of OM pyrolysis end up in the same fraction as graphitic C.

In summary, graphitic C content in soils has received very little attention as a quantification method is lacking. This study aims to test several available methods for identifying and quantifying graphitic C content of soils by examining Fourier Transform Infrared spectroscopy (FTIR), thermogravimetric analysis (TGA) and the smart combustion methods. To test the validity of the above methods for graphite identification and quantification, we analysed both natural and artificial soils that included widely present soil C components, i.e. carbonates and OM.

## 2 Material and Methods

### 2.1 Artificial mixtures and soil / rock samples

Top soil and fresh rock samples from a nearby outcrop, were taken from a field site in Rambla Honda, Sierra de los Filabres (37°7'43" N, 2°22'30" W / Southern Spain). The area is located in the Nevado-Filabride complex and contains Devonian-Carboniferous slaty micaschist with graphite and garnets crossed by abundant quartz veins (Puigdefábregas et al., 1996).
Carbonates found in the soil sample (0.18 % C) originated from pedogenesis and dust deposition as the parent rock does not contain carbonates. Additional soil material was collected from a field near the town of Alboloduy (37°4'9" N, 2°36'43" W), hereafter referred to as AB soil, with similar vegetation and (climatic) conditions. The lithology consists of feldspathic mica



schist (IGME, 1979), but without natural graphite and with a much higher $CaCO_3$ content (1.87 % C). The soil samples were dried at 40 °C and sieved to ≤ 2 mm.

Furthermore, three artificial soils were created, resembling a simplified version of a natural soil sample. For artificial soil 1 the organic horizon under deciduous tree species in Tharandter Wald (Saxony, Germany) was collected as OM substitute.

Muscovite (American Educational, PN B00657LD62), a primary mineral present in the collected rock and soil samples (IGME, 1975), was taken as mineral component and ground in an agate disc mill. Together with $CaCO_3$ (Merck, Darmstadt, PN 1.02066) and graphite standard material (Merck, Darmstadt PN 1.04206), the components were mixed in a pure quartz matrix (Merck, Darmstadt, PN 1.07536) in the ratio of 10 % Muscovite, 2.4 % OM (=1.0 % C), 2.1 % $CaCO_3$ (=0.25 % C) and 0.5 % graphite (=0.5 % C). Artificial soil 2 was made without carbonate and Artificial soil 3 without graphite, whereby concentrations

of the other components were kept the same (Table 1).

For testing and developing potential quantitative methods, the graphite standard was used as reference material and added to soil from Rambla Honda (hereafter calibration set 1) or with pure quartz (hereafter calibration set 2) as a matrix in different quantities from 0.1 to 4 % (Table 1). All samples were ground in an agate disc mill (Retsch GmbH, Haan, Germany) in order to achieve homogenisation.

Total C content of the different mixtures and samples were measured and checked using an elemental CN analyser (Vario EL Cube, Elementar, Langenselbold, Germany). By acidifying samples, using an excess of HCl, the carbonates were removed. After drying at 60 °C, the difference with and without acid treatment was measured and called the total inorganic C (TIC). TC and TIC values of the used samples are summarized in Table 1. We assumed that with dry combustion at 950 °C under pure oxygen atmosphere, all C holding components were decomposed or oxidized and therefore the total C content could

be measured. This assumption was later validated by performing TGA temperatures up to 1100 °C.

## 2.2 Fourier Transform Infrared (FTIR) spectroscopy

For transmission-FTIR analyses, 1 mg sample was mixed with 99 mg potassium bromide (KBr; Merck, Darmstadt; 3 sample replicates), finely ground in an agate mortar, and pressed to pellets. The transmission spectra were recorded in a Biorad FTS 135 spectrometer (BIO-RAD company, Cambridge, USA) as 16 co-added scans between wave number (WN) 4000 and

400 cm[-1] at a spectral resolution of 1 cm[-1]. The spectra were corrected against ambient air as background and were converted to absorption units. For DRIFT analyses, ground mixtures of calibration set 1 and 2 were poured into standard cups (three replicates) without any dilution. The DRIFT spectra (16 co-added scans, WN 4000 and 400 cm[-1], resolution 4 cm[-1]) were corrected for ambient air using a background spectrum of a gold target (99%; Infragold) and were converted to Kubelka–Munk units. All spectra were corrected for $CO_2$ absorption of the ambient air between WN 2400 and 2280 cm[-1] and smoothed (boxcar

moving average algorithm, factor for transmission spectra: 25, factor for DRIFT spectra: 15) using the software WIN-IR Pro 3.4 (Digilab, MA, USA). For each sample one mean spectrum was calculated from the spectra of three replicate spectra.

The partial least squares regression (PLSR) analyses of correlations between the transmission or DRIFT spectra and the graphite contents (0.1 - 4 %) of the samples were performed using R, Version 3.1.1 (R Core Team, 2014) with module PLS



(SIMPLS, cross-validation: leave-one-out) of Mevik et al. (2018). The number of components used in the calibration models followed the lowest predicted root-mean-square error (RMSEP) of the specific datasets. The scores and loadings were plotted for the two main components determining most of the variances of the DRIFT spectra. Larger absolute loading values of signal intensities in certain WN regions imply a greater importance of these WN for the cumulated values of the principal component 1 or 2 displayed in the score plot.

## 2.3 Thermogravimetric analysis (TGA)

The TGAs were conducted on a STA 449 F5 Jupiter analyser (NETZSCH, Hanau, Germany). Therefore, 20-40 mg of sample material was placed in an $Al_2O_3$ crucible and heated under a constant heating program from ambient to 1100 °C with a ramp of 20 °C $min^{-1}$. First analyses were conducted under an oxygen-rich atmosphere, with an inflow of 250 ml $O_2$ $min^{-1}$ and 250 ml $N_2$ $min^{-1}$. Additional tests were done under anoxic conditions whereby the oxygen inflow was cut off between 500 and 850 °C. The oxygen inflow was restored and the heating program continued until 1100 °C.

As carbonates might interfere in the TGA measurement of graphite and high chloride concentrations damages the equipment, they were removed from the sample using the acid fumigation method of Harris et al. (2001). Briefly, about 40 mg of sample was weighed in a silver foil capsules, moistened to approximately field capacity and put in a desiccator under vacuum conditions with a beaker of 31% HCl and fumigated for 24 hours. Afterwards the sample was dried at 60 °C overnight before it was transferred to an $Al_2O_3$ crucible for analysis.

TGA measurements were processed and thermal mass loss data obtained via the Proteus Thermal Analysis software (NETZSCH, Hanau, Germany). Further analyses of the obtained data were conducted using R, Version 3.5.1 (R Core Team, 2018). Using the Fitting Linear Models function, models were created from the calibrations sets. With the module PLS (cross-validation: leave-one-out) of Mevik et al. (2018), the best temperature limits for quantification of graphitic C in the calibration sets was determined, following the lowest RMSEP of the specific datasets.

## 2.4 Smart combustion

Smart combustion denotes the method based on the DIN19539 (GS) standard (DIN Standards Committee Water Practice, 2016), whereby solid C components are separated based on their thermal and oxidizable properties. Smart combustion was conducted with the Soli-TOC cube analyser (Elementar, Langenselbold, Germany). The device is equipped with a non-dispersive infrared detector (NDIR), which measures the degree of infrared light absorbance caused by $CO_2$ concentration in the measuring gas ($O_2/N_2$). The NDIR has been calibrated with $CaCO_3$ and additionally $CaCO_3$ is used to control and calculate a daily standard for the total C content measured. Depending on the expected carbonate and graphite concentration, 40-90 mg of homogenized sample was placed in the crucible. This was done to make sure that the peak surface is well within the calibration range without causing unnecessary large peak areas, which might influence the separation of the peaks / substances later on. Following the DIN19539 GS standard / standard gas switching program of the Soli-TOC cube analyser, the sample was first heated to and held at 400 °C for 240 sec. whereby the "total organic carbon 400 °C" ($TOC_{400}$) was obtained.





Subsequently the atmosphere was switched to inert gas ($N_2$) and after an equilibration time of 100 sec. the sample was heated to 900 °C and held for 150 sec. C released during this pyrolyzing phase is denoted as TIC, while it mainly consists of carbonates, which do not need oxygen to decompose. After 150 sec., the oxygen gas flow was reintroduced and a third C component, the residual oxidizable C (ROC), was measured. It was hypothesized that this ROC fraction represents graphite.

As the device directly converts the NDIR signal in the C content of the different components, as calibrated with $CaCO_3$, a Pearson correlation test was performed between the obtained ROC data and calibration sets to test how well the graphite content was measured.

## 3 Result

### 3.1 Overestimation of graphite contents by FTIR spectroscopy

The PLSR of the calibration set showed strong relations between the transmission-FTIR spectra from both calibration sets and the graphite concentrations when considering samples with $0.1 – 4$ % graphite (Figs. 1a and 1b). For DRIFT spectra, the quality of these calibrations was at the same level (cal. Set 1: $R^2 = 0.97$, RMSEP = 0.16; cal. Set 2: $R^2 = 0.98$, RMSEP = 0.12). For calibration set 1 (based on natural soil) as well as for calibration set 2 (based on quartz), one main component of the PLSR presented most differences in the graphite concentration (Figs. 1c and 1d). This component showed the highest loading values

across the entire range of wave number with some exceptions. For calibration set 1, wave numbers with decreased loading values were found at spectral regions 1077, 1031, 1013, 934, 913, 778, 536, 471, and 411 $cm^{-1}$, which all corresponded to the prominent absorption bands of the original soil used as matrix, comprising functional groups from organic and mineral matter (Hesse et al., 1984; Senesi et al., 2003; Van der Marel and Beutelspacher, 1976). For calibration set 2, the wave numbers with the smallest loadings at 1171, 1084, 796, 778, 694, 506, and 457 $cm^{-1}$ were unexceptionally specific for quartz (Van der Marel

and Beutelspacher, 1976).

The validation by the PLSR calibration of both calibration sets 1 and 2 using spectra from the other calibration set, respectively, showed linear relationships. However, this relationship produced an overestimation in absolute graphite contents of ca. 2.59 % C for set 1 and 1.87 % C for set 2 (Fig. 5). The graphite content of 0.50 % for the artificial soil was 3.5-times overestimated by the PLSR calibration using set 1 (predicted: 1.75 % C) and 3-times underestimated by calibration set 2

(predicted: 0.17 % C). The graphite contents in the graphitic schist was estimated to be 1.91 % C by calibration set 1 and 3.71 % C by calibration set 2, which is respectively 2-times and 4.5-times higher than the total C content of the graphitic schist (Table 1).

### 3.2 Strong matrix dependency of TGA predictability

First qualitative TGA results revealed overlapping mass loss peaks of graphite and $CaCO_3$ (Fig. 2). Between 750 and 850 °C,

the sum of the mass losses of the individual components was smaller than the mass loss of the mixture of these components.



Using the two calibration sets of soil and quartz with graphite, the most useful temperature range for modelling graphite content was identified. Based on the RMSEP values, the best range for modelling graphite content by mass loss was identified between 680 °C and 840 °C, as visualized in Supplementary 3. According to PLSR, both models, created with calibration set 1 ($R^2$ = 1.00, RMSEP = 0.05) and calibration set 2 ($R^2$ = 1.00, RMSEP = 0.04), the prediction of the graphite

contents in their respective matrix were quite good (Fig. 3). Only cross-validation by predicting graphite content in the other matrix revealed a short coming of the TGA method (Fig. 5). The slope between predicted and actual graphite content is still parallel to the 1:1 line, but applying the model derived from calibration set 1 on set 2 underestimated the graphite content by 1.81 % C (Fig. 5). In contrast, the graphite content in calibration set 1 was underestimated by 1.81 % C using the model calibrated with set 2.

The graphite content of the artificial soil (0.5 %) and graphitic schist were estimated using the two calibrated models (Inset Fig. 5 and Table 2). The content of artificial soil 1 was overestimated 3.5-times (predicted: 1.70 % C), using the model derived from calibration set 2. Using the model based on calibration set 1, the prediction of graphite content yielded negative values (predicted: -0.19 % C). What further stands out in Table 2 are the artificial soils 2 and 3, which were without respectively $CaCO_3$ and Graphite. Both were underestimated with the model based on calibration set 1 (based on soil) and overestimated

with the model based on calibration set 2 (based on quartz). Independent from the two models, the relative difference between them is in both cases 0.44, which resembles the actual graphite content. The graphite content of the graphitic schist was estimated to be 2.44 % C according to calibration set 2, but the model calibrated with set 1 showed better predictions. Where the total C content of the graphitic rock was 0.84 % C, the model of calibration set 1 estimated graphite content of 0.64 % C (Table 2).

Furthermore, the artificial soils were used to explore whether changing between oxic and anoxic conditions during thermal analysis could separate between the mass loss peak of $CaCO_3$ and graphite – most important for potential application in soils containing both. Changing the atmospheric composition resulted in an artificial mass gain when $O_2/N_2$ gas was switched to $N_2$ and an artificial mass loss when oxygen was reintroduced (Fig. 4), probably due to changes in pressure, and thereby affecting the mass readings. It should be noted that a constant heating program was used and therefore the measurement time

spans 4-5 minutes for both peaks. Nonetheless using artificial soil 2 (without $CaCO_3$) and artificial soil 3 (without graphite) revealed that qualitative separation between the two mass loss peaks was feasible using changes in oxic conditions. Based on TGA observations on some individual components and simplified artificial soil (Fig. 2), it seemed best to use anoxic conditions from 500 till 850 °C as all OM will be oxidized at 500 °C and carbonates should be fully decomposed at 850 °C (Fig. 2). Furthermore, the mass loss peak had already returned to the baseline after reaching 850 °C, indicating that the decomposition

of the carbonates was completed (Fig. 6).

## 3.3 Direct graphite content quantified by smart combustion

First total carbon (TC) measured by the smart combustion method was compared with the TC obtained by dry combustion using the elemental analyser, but hardly any differences were found (Supp. 4). Residual oxidizable C (ROC) values obtained



by the smart combustion method were plotted against the added amount of graphite in calibration sets (Fig. 5). The graphite content in calibration set 1 seems to be overestimated by 0.26 % C. This observation can be explained by the fact that the used sample soil for calibration set 1 contains an unknown amount of natural graphite, which explains a constant overestimation. The content of graphitic C of the soils in calibration set 2 is slightly underestimated, especially with higher graphite concentrations (Fig. 5 and Table 2).

It can be seen in Table 2 that the graphite content of the artificial soil 1 was slightly underestimated: 0.40 % ROC for the artificial soil 1 and 0.46 % ROC for the artificial soil 2, where it should have been 0.50 %. Artificial soil 3 measured 0.00 % ROC as there was also no graphite in this sample. The graphitic schist had 0.79 % ROC, which was very similar to the total C of the rock (Table 1 and 2). Furthermore, an additional sample without natural graphite but with a high $CaCO_3$ content (1.87 % C), the AB soil spiked with graphite, showed a similar underestimation as observed with the calibration samples (Fig. 5).

The natural graphitic soil, used for calibration set 1, was also spiked with $CaCO_3$, for which the ROC results are given in Figure 6. Although dilution of the sample by the addition of $CaCO_3$ was taken into account, a downward trend of measured ROC content of 0.01 % ROC absolute was observed with increasing amounts of added $CaCO_3$ (from 0.0 to 2.5% added C) explaining at least parts of the underestimation of graphite in this sample (Fig. 5).

**4 Discussion**

**4.1 Matrix effects and the lack in specific absorption bands hamper graphite quantification via FTIR spectroscopy**

The calibration between infrared spectra and graphite contents of the calibration sets yielded promising results (Figs. 1a and 1b) and could also be used for a cross-validation (Fig. 5). However, these validations showed a systematic overestimation of graphite contents for both calibrations and the inability of these calibration sets for predicting the graphite content of artificial

soil 1.

The loading values revealed that the relationships used for the PLSR calibration (Figs. 1c and 1d) could not be attributed to absorption bands of graphite reported in literature such as wave numbers 2200, 1587, 1362, and 830 cm[-1] (Friedel and Carson, 1971) or 3400, 1729, 1220, 1052 and 870 cm[-1] (Tan et al., 2013). These wave number positions did not match with the absorption bands of the spectra obtained from calibration set 1 and 2, the graphitic schist and the artificial soil 1 (Fig.

7). The only exception is WN 3400 cm[-1], which is mainly caused by O–H of free or adsorbed water and is thus not specific for graphite.

The height of the loading values across broad spectral regions, i.e. across the entire wave number range in case of the soil samples, hints on effects of the general optical conditions within the samples. The transmission, i.e. the energy throughput in the sample pellet (transmission FTIR) or the reflectance of sample surface (DRIFT), seems to be a measure for the amount

of graphite added rather than specific graphite signal intensities in the calibration sets, but transmission/reflection characteristics are highly influenced by the mineral composition of a sample. Generally, increasing graphite concentrations caused decreasing transmission over the entire spectral range, which is a characteristic of the mineral composition, due to



increasing proportions of primary absorption (Kortüm, 1969; Hesse et al., 1984). This assumption is underlined by the fact that calibration was not possible with both calibration sets when using baseline-corrected spectra, because baseline correction compensated the described effect of decreasing transmission with increasing graphite content. In addition, DRIFT measurements of pure, i.e. undiluted, graphite material (not shown) did not reveal any prominent absorption bands. Note that

specific graphite absorption bands that have been reported in the literature are only valid for oxidized graphite where C–O and C=O groups have been formed to a certain extent (Tan et al., 2013). While FTIR spectroscopy may be feasible for determining oxidized or impure graphite, it was found not useful in our case, where an oxidization or impureness of the used graphite material obviously did not appear. Further, the mentioned potential signal intensities would occur in the same spectral ranges as compared to signals from SOM functional groups, thus hampering a quantification of graphite in soil samples. Consequently,

the lack of specific absorption bands resulted in a strong dependency of the calibration and validation quality on the sample matrix, i.e. its main mineral component. This matrix effect was illustrated by the incapability of the PLSR models to predict the graphite contents of the artificial soil 1 or graphitic schist (Fig. 5 and Table 2).

### 4.2 Strong matrix effects did not allow using TGA as a universal quantification method

In recent work, the TGA method has been tested and further developed for differentiating between carbonates and OC / OM

(e.g. Apesteguia et al., 2018). First analysis of individual components in a quartz matrix revealed that graphite has a similar thermal stability as carbonates (Fig. 2) and overlap the thermal region where dehydroxylation of various minerals takes place (Földvári, 2011; Fernández et al., 2012). As shown in supp. 4 and discussed by other studies, complete removal of carbonates from soil sample by acid fumigation is difficult and quantitative estimation, as sought for graphite, via TGA becomes challenging as the acid affects the thermal stability of other soil constituents and makes the sample hygroscopic (Agarwal and

Bucheli, 2011; Apesteguia et al., 2018). Additionally, sample grinding in an agate disk mill, representing common homogenization process used for small sample amounts (10's of mg), introduce some changes in thermogravimetric patterns for some minerals, e.g. micas, but makes it also more "reactive" (Földvári, 2011). This would mean that mass loss peaks for minerals, like the used muscovite, can appear sharper and at lower temperatures, in comparison with non-ground materials.

       The best temperature ranges to relate mass loss to the amount of added graphite was between 680 °C and 840 °C. For

calibration set 2 (pure quartz matrix) a lower temperature (range) would also be able to predict the graphite content (Supp. 3), which indicates an interference in the soil matrix of calibration set 1. The best temperature range was in line with the observation that the mass loss peak of graphite spans a large range (Fig. 2), most likely a result of the slow oxidation of this pure C. Other studies found that graphite in a (fluid) sand bed already oxidized slowly under oxygen rich conditions at temperatures below 670 °C accelerating at higher temperatures (Hayhurst et al., 1998; Bews et al., 2001).

30         Validation of the created models from the two calibration sets revealed that interference with other soil components required an individual calibration for every sample set of specific (mineral) composition (Fig. 5). As shown in this study, fresh but ground muscovite dehydroxylates between 600 and 1000 °C (Fig. 2), which influenced the total mass loss measured in this temperature range. Other present (minor) minerals, like chlorites (500-860 °C) or apatites (200-1400 °C), might also increase





the bias by influencing mass loss (Földvári, 2011; Tõnsuaadu et al., 2011). This observation could explain why the model of calibration set 1, using the soil spiked with graphite, showed a good predictability of the graphite content in the graphitic schist, as mineral composition is highly similar between these two samples (Fig. 5 and Table 2).

5       Roth et al. (2012) suggested that the use of anoxic conditions / a pyrolyzing phase during measurement might be useful to differentiate between wood and black C. According to our gas switching experiment with the TGA, it is at least a useful approach to differentiate between graphite and $CaCO_3$ (Fig. 4). Due to the artificial mass gain/loss induced by switching the gases during the measurement, exact temperature ranges for developing a quantitative method could not be established. As no universally-accepted temperature limits for the quantification of TOC, TIC or other carbonaceous substances exist, the best temperature ranges for switching between oxic and anoxic conditions are difficult to define (Pallasser et al., 2013; Ussiri et al.,

2014). For instance, according to the DIN19539 - standard, TOC is defined as the oxidizable C at maximum of 400 °C. Others showed that 1/5 or even 1/3 of the TOC is not oxidized at 400 °C (Pallasser et al., 2013; Schiedung et al., 2017). For the artificial soil in our study a temperature limit of 400 °C seems to be too low to oxidize all OM, as indicated by the TGA in Fig. 2, and therefore the pyrolyzing phase was set to 500 °C. To obtain a clear peak for the graphite oxidation, it is important that the other substances, i.e. $CaCO_3$, are already decomposed. In the case of the artificial soil, it was found that at 850 °C all $CaCO_3$

was decomposed and a clear peak for graphite was formed upon re-establishing the oxic conditions (Fig. 4). With higher $CaCO_3$ levels or dolomitic carbonates, a higher temperature might be needed to create a clear separation between the substances (Földvári, 2011).

      TGA seemed to be a good method to identify different organic components of samples and thus can be used as complemental technique to other methods for (organic) C content estimation. For high graphite content with negligible

amounts of dehydroxylating minerals and/or decomposing carbonates, TGA might be a useful method to quantify graphite.

## 4.3 Minor effect by $CaCO_3$ and radicals on direct graphite quantification using smart combustion

With the TGA method it was already shown that qualitative differentiation between carbonates and graphite was possible by changing between oxic and anoxic conditions during heating of the sample (Fig. 4). Using the Soli-TOC device, a direct measurement of the released C could be achieved during the heating/gas changing program, which correspond very closely to

the amount of (added) graphite (Fig. 5 and Table 2). The fact that the Soli-TOC device measured almost the same TC values as the elemental analyser (Supp. 4), supported the idea that a direct comparison between the ROC fraction and (added) graphite content is possible.

      As shown by Hayhurst and Parmar (1998), very small impurities in the graphite can cause a small part of the graphite to pyrolyse during anoxic conditions at higher temperatures. Taking a closer look at the measurements of the artificial soil,

reveals that a small part of the graphite started to oxidize under anoxic conditions (Fig. 8). Bews et al. (2001) suggested that at temperatures higher than 700 °C, radicals like $HO_2$ and $OH$ might act as reactant with the pure C. Furthermore, in the method comparison study for recovering different black C types, Roth et al. (2012) suggested a (relatively) strong catalytic effect of oxides on black C oxidation, which was most predominant in soils. These ideas are also supported by our observation that





artificial soil 2 (without $CaCO_3$) measured higher ROC values (0.06% more C absolute) than artificial soil 1 (with $CaCO_3$, Table2). Also in the carbonate-rich AB soil the added graphite was underestimated by 7% (Fig. 5). Furthermore, graphitic C was underestimated with increasing $CaCO_3$ content (Fig. 6). The 7 % underestimation by the AB soil, which contains 1.87 % C-$CaCO_3$, coincided with ROC underestimation of the calibration soil with 2 %C-$CaCO_3$ addition.

5         When the soil contains more thermally resistant OM, which is not oxidizable at 400 °C and can be 1/4[th] of the OM (Schiedung et al., 2017), the question rises if this fraction is pyrolyzed with heating under anoxic conditions or if it is taken as part of the ROC fraction when oxygen is again available. The TGA method showed that not all OM has been oxidized at 400 °C (Fig. 2). Taking a closer look on the smart combustion measurement of the artificial soil 2 and only its fresh OM component (Fig. 8), revealed a small peak formed upon heating the sample above 400 °C, which is only a few percent of the total OC, but

a clear indication that not 100% of the OM was oxidized at 400 °C. Although this study focuses on the ROC component, it might be important when considering the $TOC_{400}$ and TIC fractions of the smart combustion method.

        Although the thermal boundaries for the different C fractions are given in the DIN19539 (GS) standard are debatable (Ussiri et al., 2014; Schiedung et al., 2017), we showed that the ROC fraction corresponded closely to the graphitic C content. Through the smart combustion method, graphitic C could be differentiated from the other C components in soil matrix and

quantified satisfactory as indicated by the offset in the calibration with graphite estimation in the artificial/spiked samples.

### 4.4 Potential for combining methods

Comparing the ability of the examined methods on predicting graphite content, it becomes clear that FTIR overestimated, TGA was highly variable, and smart combustion was most accurate in predicting the graphite content (Fig. 5). An interesting observation was the similar predicted graphite content in calibration set 1 by both the FTIR and TGA methods, especially as

FTIR is based on spectral properties and TGA on the thermal stability of the graphite. It has previously been suggested to combine FTIR and TGA systems to rapidly characterize the soil OM (Demyan et al., 2013). Oxidation of graphite upon heating could result in specific infrared bands (Tan et al., 2013), which nevertheless would still be superimposed by SOM-specific bands in natural soil samples. As discussed by Demyan et al. (2013), not only the available oxygen, but also the heating rate has an important effect on the charring of OM and thereby on the thermal and spectral properties of the studied material.

25        Although we focused in this study on the ROC component, which significantly correlated with the graphite content, considering the other components in the DIN19539-standard was beyond the scope of this study. Nonetheless, we found indications that the thermal boundaries defined in the DIN19539-standard are not ideal to differentiate between soil OM and inorganic C (Fig. 8). As most carbonates start to decompose at temperatures of 550 °C (Földvári, 2011), it might be more suitable to increase the level for the TOC component from 400 to 500 °C. Using TGA simultaneously with differential scanning

calorimetry, water and $CO_2$ flux measurements as previously suggested by Fernández et al. (2012), could improve the development for a more standardized method applicable to soils using combustion elemental analysers. The overlap between

the thermal properties of different C components emphasizes the need to always first consider what is present in the sample and what might interfere with the considered applied methodology.

## 5 Conclusion

Three widely used methods were examined for their potential to quantify graphitic C content in soil samples. Calibrations between mid-infrared transmission as well as DRIFT spectra and graphite contents of well-defined samples are principally possible via PLSR. However, these calibrations depend on general effects of graphite contents on the energy transmitted through the samples rather than on signal intensities of specific graphite absorption bands. The use of samples from different origin yields strong matrix effects and hampers the prediction of geogenic graphite contents in soils. Thermogravimetric analysis of the samples revealed that it is a useful qualitative method of identifying graphitic C in soil samples, although care should be taken for carbonates as they have a similar thermal stability. Quantitative estimation of the graphite content seems challenging as dihydroxylation of several soil minerals occur at similar temperatures, making the calibration with an empirical model necessary. With alteration between oxic and anoxic conditions during heating of a sample, a differentiation between other soil components and graphite could be established using smart combustion. Further quantification of the released C during the gas changing heating program, revealed a close correspondence between the measured ROC and original graphite content. Of the examined methods, the smart combustion method performs best in differentiating between graphite and other soil components and thereby also in quantifying graphitic C in soil samples.

**Author contribution**

ML contributed the FTIR part to the article, including analysis of the data, and editing the paper.

CV contributed by suggesting and evaluating test set-up and by commenting on and editing the paper.

SS contributed by providing measurements with the Soli-TOC device, technical assistance and editing the paper.

KK contributed by suggesting and evaluating test set-up and by commenting on and editing the paper.

**Competing interests**

The authors declare that they have no conflict of interest.

**Acknowledgements**

We thank Manuela Unger and Gisela Ciesielski for the laboratory assistance and the (anonymous) reviewers for their helpful input. This study was financially supported by the Deutsche Forschungsgemeinschaft (DFG), Bonn, Germany, under grants KA1737/13-1 "Extracellular polymeric substances and aggregate stability - how microorganisms affect soil erosion by water"




and LE 3177/1-2 "Quantification of small-scale physicochemical and microbiological properties of intact macropore surfaces in structured soils", and by European Research Council (ERC) under the European Union's Horizon 2020 programme, grant 695101 "Radiocarbon constraints for models of C cycling in terrestrial ecosystems: from process understanding to global benchmarking".

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





**Figure captions**

**Figure 1: Prediction plots with 95% prediction bands (a, b) and loading plots (c, d) after FTIR analyses of the PLSR calibration sets using soil and pure quartz with graphite concentrations of 0.1 – 4 % added as indicated in table 1.**

**Figure 2: Thermogravimetric analysis of artificial soil 1 and its components measured individually. The summation (dash-dotted, grey) is the combined mass loss of the individual components.**

**Figure 3: Prediction plots after thermogravimetric analysis (TGA) of calibration set 1 (squares) and 2 (diamonds). 95% predictions band (dotted grey) are displayed besides the linear regression line (black). Graphite was added in the concentrations between 0% and 4.0% graphite as indicated in table 1.**

**Figure 4: Thermogravimetric analysis of the artificial soils, with one lacking either carbonate (Artificial soil 2, green) or graphite (Artificial soil 3, orange), whereby the oxygen gas supply was cut off during part of the standard heating program (GC – Gas Change). For comparison the artificial soil 1 under normal program (without gas change) is also displayed in grey.**

**Figure 5: Overview of the predicted amount of graphite in the calibration sets (squares/diamonds), artificial soil (inset, circles/triangle), graphitic schist (inset, stars) and AB soil (right pointing triangle) as measured with the different methods. Exact data is given in Table 2.**

**Figure 6: Residual oxidizable carbon (ROC) as measured with the smart combustion method plotted against the added CaCO$_3$ content to soil sample, used for creating calibration set 1.**

**Figure 7: Transmission spectra of the pure quartz (Calibration set 2, sample 1), quartz + 4 % graphite (Calibration set 2, sample 10), soil + 4 % graphite (Calibration set 1, sample 10), artificial soil 1 (0.5 % graphite added) and graphitic schist. The vertical lines denote wave numbers for which absorption peaks have been reported in literature (see text).**

**Figure 8: Examples of smart combustion measurements of the artificial soils (a) and the fresh OM component (b). The blue area delineates the part where O$_2$ is substituted for He and the temperature program is displayed by the red dashed line. Note that artificial soil 2 (green) is without CaCO$_3$ and artifical soil 3 (orange) is without graphite.**








**Figure 1: Prediction plots with 95% prediction bands (a, b) and loading plots (c, d) after FTIR analyses of the PLSR calibration sets using soil and pure quartz with graphite concentrations of 0.1 – 4 % added as indicated in table 1.**



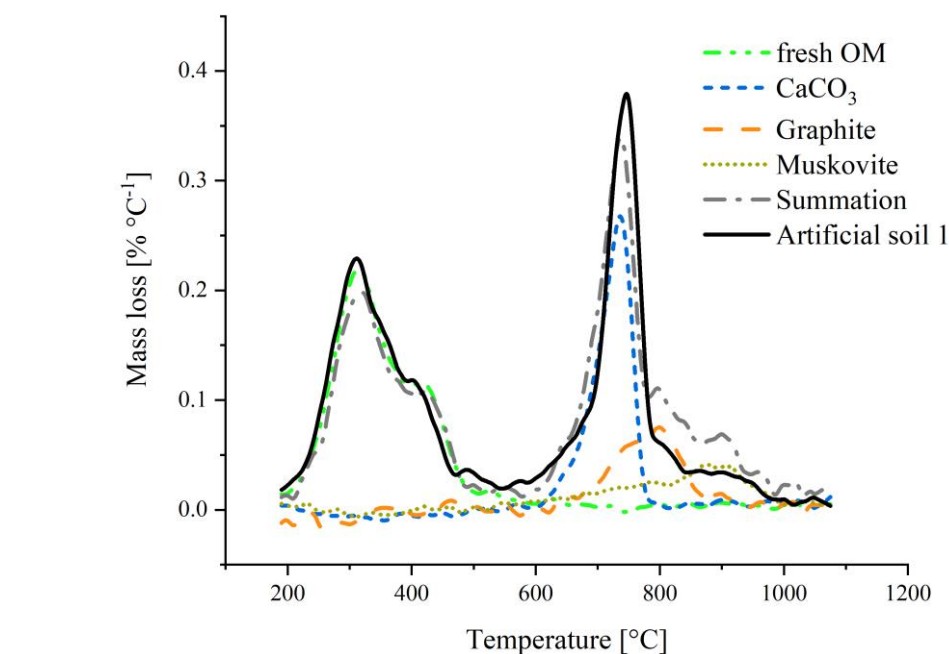

**Figure 2: Thermogravimetric analysis of artificial soil 1 and its components measured individually. The summation (dash-dotted, grey) is the combined mass loss of the individual components.**





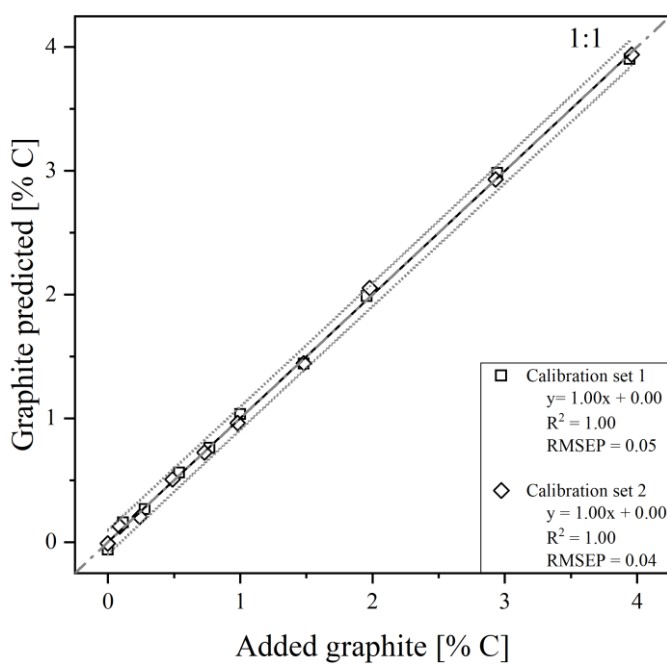

**Figure 3: Prediction plots after thermogravimetric analysis (TGA) of calibration set 1 (squares) and 2 (diamonds). 95% predictions band (dotted grey) are displayed besides the linear regression line (black). Graphite was added in the concentrations between 0% and 4.0% graphite as indicated in table 1.**





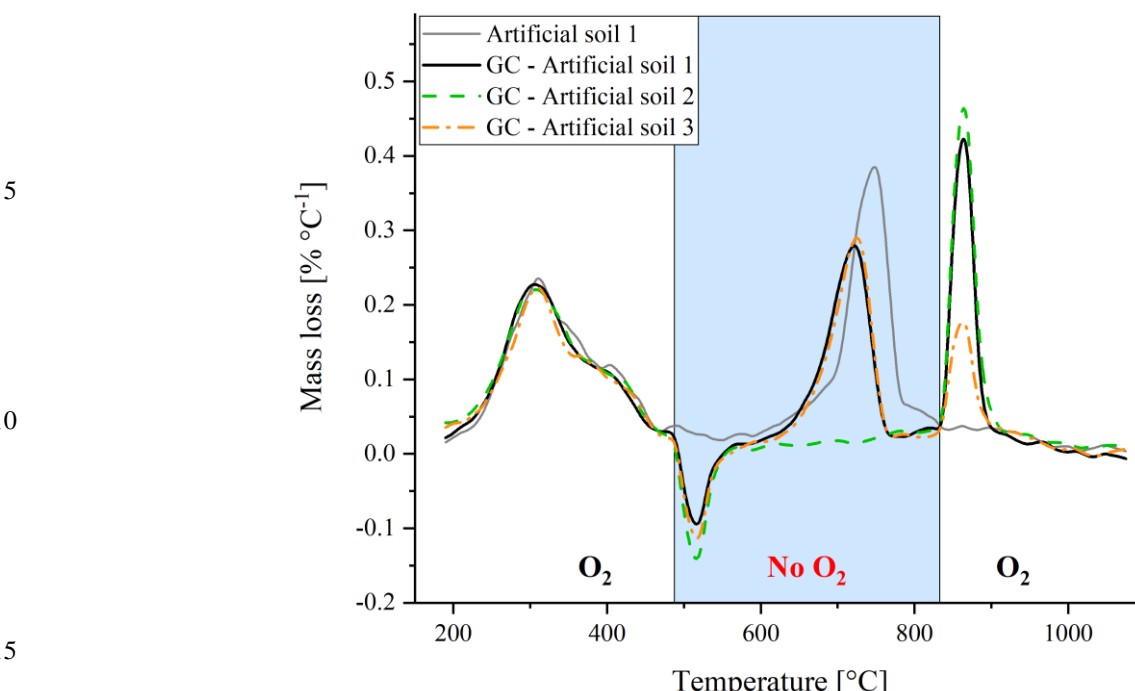

**Figure 4: Thermogravimetric analysis of the artificial soils, with one lacking either carbonate (Artificial soil 2, green) or graphite (Artificial soil 3, orange), whereby the oxygen gas supply was cut off during part of the standard heating program (GC – Gas Change). For comparison the artificial soil 1 under normal program (without gas change) is also displayed in grey.**



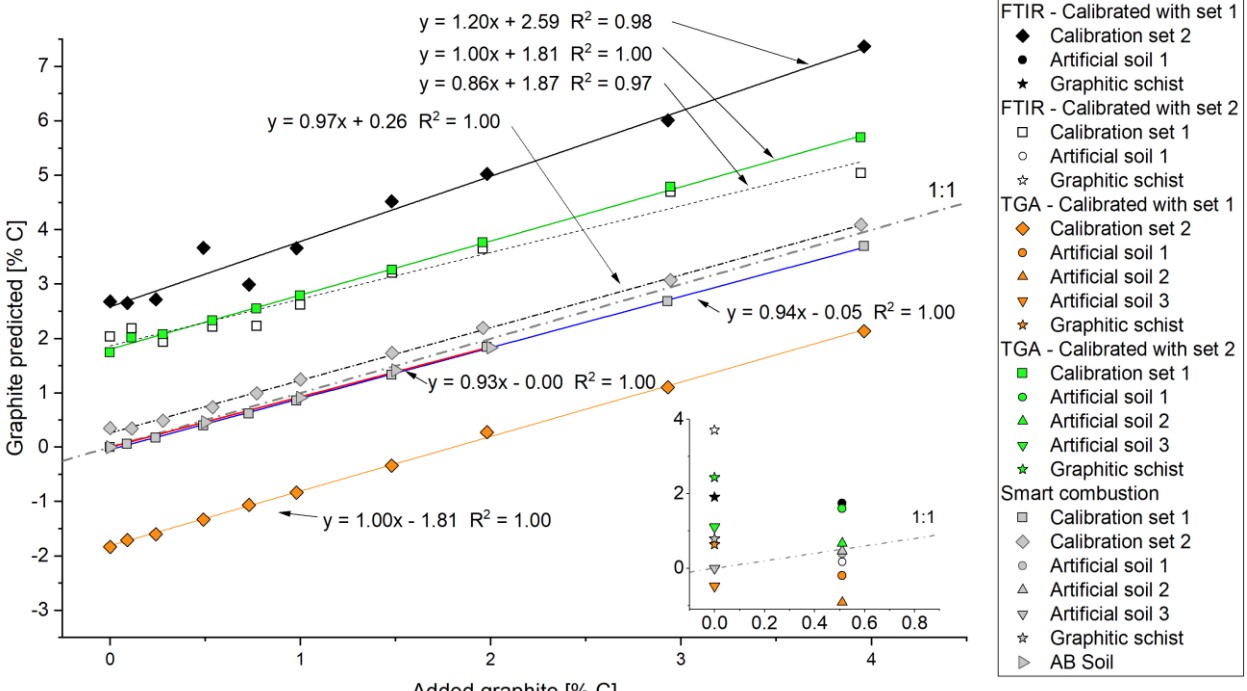

**Figure 5: Overview of the predicted amount of graphite in the calibration sets (squares/diamonds), artificial soil (inset, circles/triangle), graphitic schist (inset, stars) and AB soil (right pointing triangle) as measured with the different methods. Exact data is given in Table 2.**





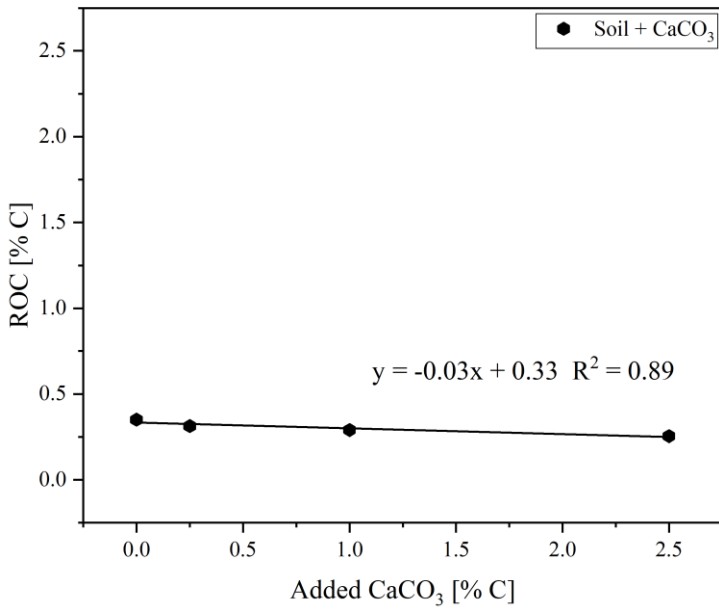

**Figure 6: Residual oxidizable carbon (ROC) as measured with the smart combustion method plotted against the added CaCO₃ content to soil sample, used for creating calibration set 1.**





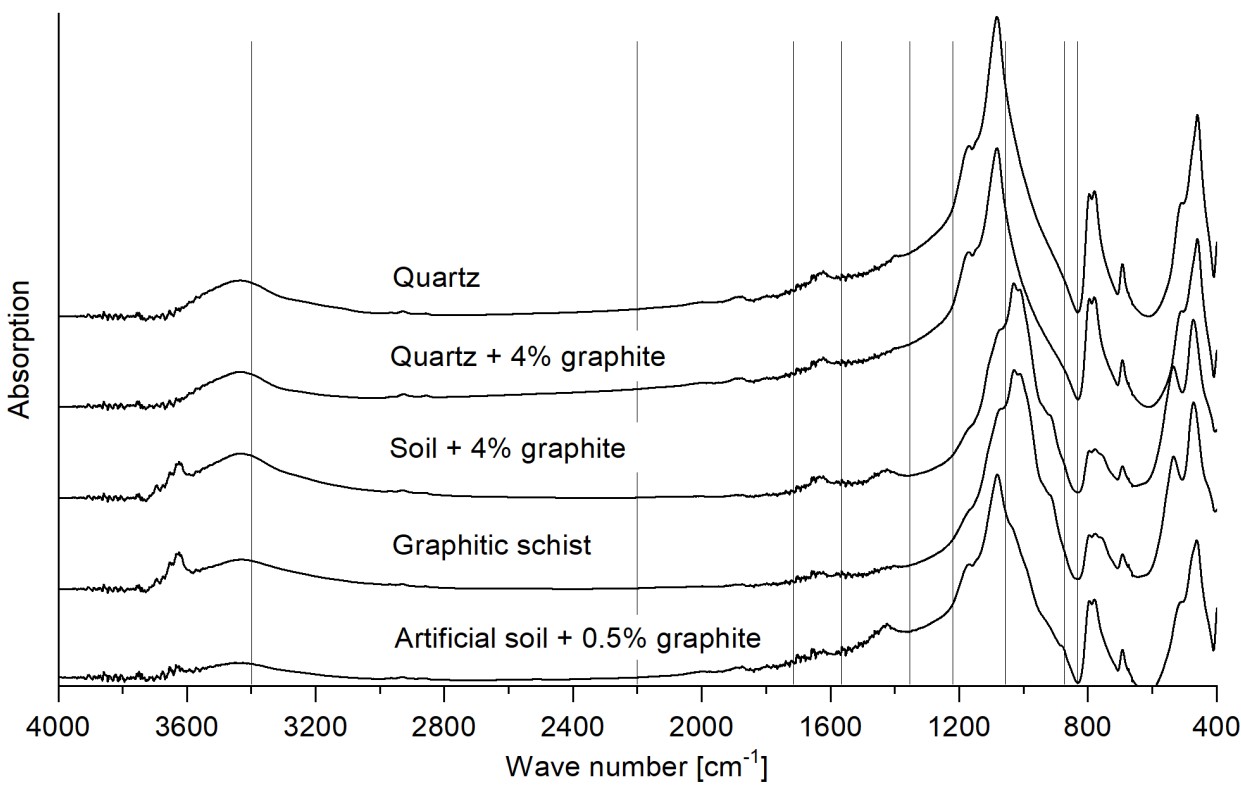

**Figure 7: Transmission spectra of the pure quartz (Calibration set 2, sample 1), quartz + 4 % graphite (Calibration set 2, sample 10), soil + 4 % graphite (Calibration set 1, sample 10), artificial soil 1 (0.5 % graphite added) and graphitic schist. The vertical lines denote wave numbers for which absorption peaks have been reported in literature (see text).**





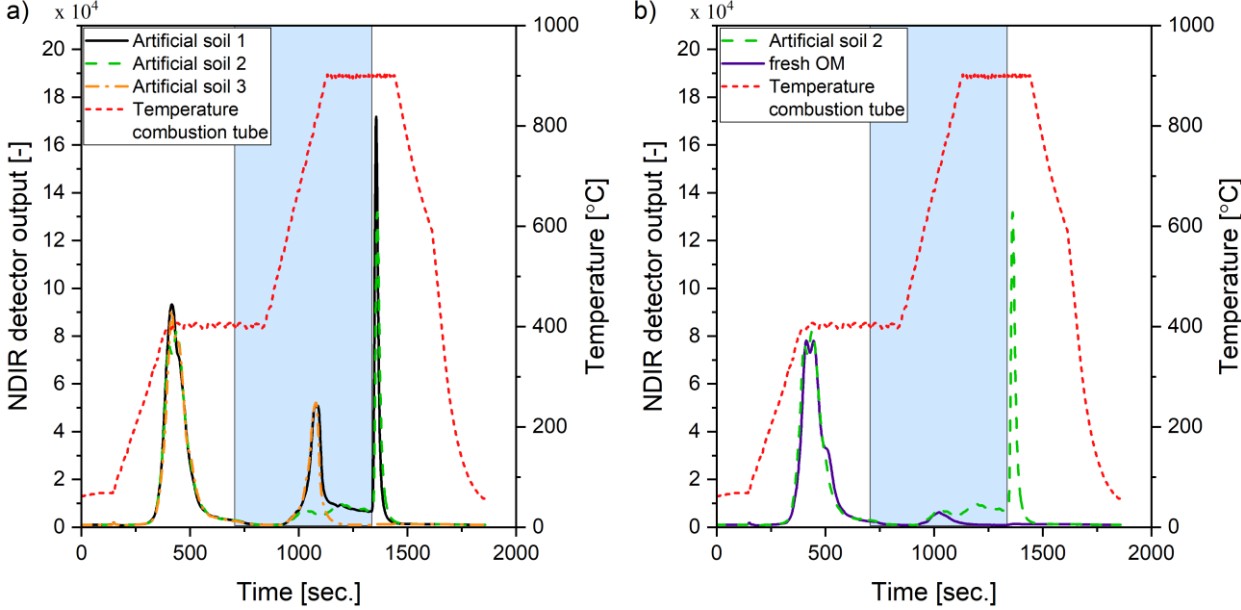

**Figure 8: Examples of smart combustion measurements of the artificial soils (a) and the fresh OM component (b). The blue area delineates the part where O₂ is substituted for He and the temperature program is displayed by the red dashed line. Note that artificial soil 2 (green) is without CaCO₃ and artifical soil 3 (orange) is without graphite.**

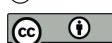



**Table 1: Overview of the used samples. Total Carbon (TC) and Total Inorganic Carbon (TIC) data is derived by the elemental analyzer, as described in section 2.1. Crosses note for which method testing the sample has been used.**

| Sample | TC [% C] | TIC [% C] | Added Graphite [% C] | *FTIR* | *TGA* | *Smart combustion* |
|---|---|---|---|---|---|---|
| Calibration 1(1)* | 1.36 | 0.18 | 0.0 | X | X | X |
| Calibration 1(2)* | 1.47 | 0.18# | 0.1 | X | X | X |
| Calibration 1(3)* | 1.64 | 0.18# | 0.25 | X | X | X |
| Calibration 1(4)* | 1.90 | 0.18# | 0.5 | X | X | X |
| Calibration 1(5)* | 2.13 | 0.18# | 0.75 | X | X | X |
| Calibration 1(6)* | 2.36 | 0.18# | 1.0 | X | X | X |
| Calibration 1(7)* | 2.84 | 0.18# | 1.5 | X | X | X |
| Calibration 1(8)* | 3.32 | 0.18# | 2.0 | X | X | X |
| Calibration 1(9)* | 4.30 | 0.18# | 3.0 | X | X | X |
| Calibration 1(10)* | 5.30 | 0.18# | 4.0 | X | X | X |
| Calibration 2(1) | 0.01 | 0.00 | 0.0 | X | X | X |
| Calibration 2(2) | 0.10 | 0.00# | 0.1 | X | X | X |
| Calibration 2(3) | 0.25 | 0.00# | 0.25 | X | X | X |
| Calibration 2(4) | 0.50 | 0.00# | 0.5 | X | X | X |
| Calibration 2(5) | 0.74 | 0.00# | 0.75 | X | X | X |
| Calibration 2(6) | 0.99 | 0.00# | 1.0 | X | X | X |
| Calibration 2(7) | 1.49 | 0.00# | 1.5 | X | X | X |
| Calibration 2(8) | 1.99 | 0.00# | 2.0 | X | X | X |
| Calibration 2(9) | 2.94 | 0.00# | 3.0 | X | X | X |
| Calibration 2(10) | 3.97 | 0.00# | 4.0 | X | X | X |
| Artificial soil 1 | 1.74 | 0.30 | 0.5 | X | X | X |
| Artificial soil 2 | 1.46 | 0.00 | 0.5 | | X | X |
| Artificial soil 3 | 1.21 | 0.29 | - | | X | X |
| Graphitic schist* | 0.84 | 0.00 | - | X | X | X |
| AB soil 1 | 2.97 | 1.87 | - | | | X |
| AB soil 2 | 3.62 | 1.87# | 0.5 | | | X |
| AB soil 3 | 4.05 | 1.87# | 1.0 | | | X |
| AB soil 4 | 4.51 | 1.87# | 1.5 | | | X |
| AB soil 5 | 4.97 | 1.87# | 2.0 | | | X |

*\* -contain unknown amount of natural graphite // # -as measured in the sample with 0.0 added graphite*



**Table 2: Overview of the predicted graphitic carbon by the different examined methods.**

| Sample | Added Graphite | FTIR | | TGA | | Smart combustion |
| --- | --- | --- | --- | --- | --- | --- |
| | | *Calibrated with Cal. Set 1* | *Calibrated with Cal. Set 2* | *Calibrated with Cal. Set 1* | *Calibrated with Cal. Set 2* | |
| | [% C] | [% C] | [% C] | [% C] | [% C] | [% C] |
| Calibration 1(1)* | 0.0 | - | 2.04 | -0.06 | 1.54 | 0.35 |
| Calibration 1(2)* | 0.1 | 0.29 | 2.19 | 0.16 | 1.76 | 0.34 |
| Calibration 1(3)* | 0.25 | 0.20 | 1.94 | 0.27 | 1.87 | 0.49 |
| Calibration 1(4)* | 0.5 | 0.84 | 2.22 | 0.56 | 2.17 | 0.74 |
| Calibration 1(5)* | 0.75 | 0.53 | 2.23 | 0.76 | 2.37 | 0.99 |
| Calibration 1(6)* | 1.0 | 0.79 | 2.62 | 1.03 | 2.65 | 1.25 |
| Calibration 1(7)* | 1.5 | 1.39 | 3.21 | 1.44 | 3.06 | 1.74 |
| Calibration 1(8)* | 2.0 | 2.15 | 3.66 | 1.99 | 3.61 | 2.20 |
| Calibration 1(9)* | 3.0 | 3.30 | 4.70 | 2.98 | 4.62 | 3.08 |
| Calibration 1(10)* | 4.0 | 3.61 | 5.04 | 3.90 | 5.55 | 4.09 |
| Calibration 2(1) | 0.0 | 2.68 | - | -1.59 | -0.01 | 0.00 |
| Calibration 2(2) | 0.1 | 2.66 | 0.26 | -1.46 | 0.13 | 0.06 |
| Calibration 2(3) | 0.25 | 2.72 | 0.25 | -1.38 | 0.20 | 0.17 |
| Calibration 2(4) | 0.5 | 3.67 | 0.58 | -1.08 | 0.51 | 0.39 |
| Calibration 2(5) | 0.75 | 2.99 | 0.67 | -0.87 | 0.72 | 0.61 |
| Calibration 2(6) | 1.0 | 3.66 | 0.85 | -0.63 | 0.96 | 0.85 |
| Calibration 2(7) | 1.5 | 4.52 | 1.54 | -0.15 | 1.45 | 1.33 |
| Calibration 2(8) | 2.0 | 5.03 | 2.06 | 0.45 | 2.05 | 1.84 |
| Calibration 2(9) | 3.0 | 6.01 | 2.89 | 1.31 | 2.93 | 2.68 |
| Calibration 2(10) | 4.0 | 7.36 | 4.14 | 2.31 | 3.94 | 3.70 |
| Artificial soil 1 | 0.5 | 1.75 | 0.17 | 0.09 | 1.70 | 0.40 |
| Artificial soil 2 | 0.5 | - | - | -0.92 | 0.67 | 0.46 |
| Artificial soil 3 | 0.0 | - | - | -0.48 | 1.11 | 0.00 |
| Graphitic schist* | - | 1.91 | 3.71 | 0.62 | 2.23 | 0.79 |
| AB soil 1 | - | - | - | - | - | 0.00 |
| AB soil 2 | 0.5 | - | - | - | - | 0.46 |
| AB soil 3 | 1.0 | - | - | - | - | 0.91 |
| AB soil 4 | 1.5 | - | - | - | - | 1.42 |
| AB soil 5 | 2.0 | - | - | - | - | 1.83 |

\* -contain unknown amount of natural graphite