# Peer review of "Identifying and quantifying geogenic organic carbon in soils – the case of graphite"

_SOIL, 2019_

## Referee Comment (RC1) · Anonymous Referee #1 · 14 Jul 2019

I have read with interest the draft untitled "Identifying and quantifying geogenic organic carbon in soils – the case of graphte". Overall, I have found that the draft is very clear. To be published in SOIL, I consider that the authors should provide a proper description of the soils they used in the study. I have also several (rather minor) concerns and questions that have to be answered before I can recommend the publication of this draft in SOIL.

1° Graphitic C can be found in rocks. Depending on P and T conditions experienced by the sediments, we do not get necessarily pure graphite. I am therefore wondering if the graphite standard material (Merck) is similar to the graphite found in the soils developed in micaschists. I would appreciate to see Raman signatures of the graphites used in the study. We can also imagine that some graphitic C with lots of defaults may evolved

before the final oxidation step. In this case, such graphitic C would not be recovered in the ROC fraction. Can the authors discuss or rule out this hypothesis?

2° Pyrogenic C (pyOC) can also resist to high T under anoxic conditions. In this case, some pyOC may be recovered in the ROC fraction. What would happen if the studied soil contains both graphite and pyOC? It may have been interesting to add charcoal in the tested mixtures. If the presence of pyOC is a limit to the method, it should be discussed.

3° 10 samples for calibrating a model is definitely a too low number of samples. It would have been highly surprising to get nice results with such a low number of samples. It may have been interesting to use all the samples to design calibration models. We can't exclude that with a nice sample set containing 500 samples with known graphite concentrations, a convincing IR-based model can be designed.

4° The authors hypothesized that ROC content would match graphite content. It is not too far but not perfect. Why don't the authors try to design a model based on ROC results as they did with IR and TGA results?

5° I do not understand the Figure 5. I suggest improving the explanations on this Figure or removing it

---

## Referee Comment (RC2) · Anonymous Referee #2 · 23 Jul 2019

This is a very well written manuscript that reports a methods comparison for the identification/quantification of graphite in soil. The writing is excellent, and the reporting of results is clear. But the study have a few important shortcomings that I'd like to see addressed before I would deem it acceptable for publication.

1. The manuscript does not provide sufficient context for focusing exclusively on graphite. The reader might interpret the current rationale as a narrow justification for using graphite in the experiment. I understand that graphite may form from metamorphic processes, but is the goal of the study to quantify graphite specifically, or geogenic organic C more broadly? The former seems far too narrow a prospect given the wide range of forms of geogenic C, and the latter is underdeveloped in the study. Without further elaboration, graphite seems a little too specific. Galy, Hemingway and others

are not specific when they refer to "lithospheric" C, Ussiri (cited in the manuscript), Chan (2017, Themochimica Acta) and others have targeted coal, and there is a large and growing literature on pyrogenic C in soils. How would the presence of these affect results? Can graphite be distinguished from other forms of thermally recalcitrant organic C? The authors do well to distinguish between carbonates (which varies widely among dolomite, calcite, etc.) and thermally recalcitrant C, but have not adequately elaborated on graphite versus other forms of geogenic C, let alone pyrogenic C. The distinction between the latter two is obvious using 14C, but the issue here is among geogenic C forms.

2. As a result of the discussion above, and more generally for a method development study, I found the number and range of materials used too small. Only one graphite-containing natural soil and one carbonate-containing natural soil were used for validation. The artificial soil mixtures were made from one OM source far removed from the natural soil (Germany vs. Spain), and "neat" mineral specimens (quartz, muscovite, CaCO3 (not dolomite or calcite?)). While I understand the desire to create a reductionist, simplified system for initial testing, the result is only nine samples of one mixed-matrix to generate the calibration. This is a highly undersampled relationship. This is critically important because the authors are right to highlight matrix effects, but they do not adequately account for these in the design of the calibration/correlation study. CaCO3 is not dolomite or calcite, soil minerals are often interstratified, and graphite likely exists in a mineral-associated form fused to the mineral matrix. None of these incipient properties of soil are accounted for in the method development – making the results of limited value. It is a nice proof of principle, but the study needs to go well beyond this given the current state of the literature. I don't see how this study substantially move us further than some of the studies cited within it.

3. The authors identified one of the challenges of thermally distinguishing forms of C as the determination of threshold temperatures. However, the description of how TGA data was processed is inadequate. Phrases like "models were created from the

calibrations sets" and "the best temperature limits for quantification of graphitic C in the calibration sets was determined" are not reproducible. These steps may be the most critical step of the process, but even the most experienced expert in this field would be unable to verify and repeat it. More detail is required here. There are also no details provided on how these data were used in the calibration. The selection of different threshold temperatures somewhat undermines the "smart combustion" approach if it cannot be universally applied. Perhaps there is some elaboration required in the discussion about how much control over threshold temperature there is available with such an instrument, and whether the DIN methods are suitable/adequate standards.

4. I would argue that since this is such a key/core component of the study, that it deserves its own separate subsection within the methods (ie, statistical analyses). I'm not 100% sure that my comments here will be relevant or correct because it was difficult to follow precisely how each of the calibrations were generated. But if I read it correctly, the authors appear to use different methods to generate the calibration curves depending on the quantification method. PLS was used for the FTIR method of quantification, not sure how it was done for what data from TGA, and Pearson correlation for the "smart combustion" data. In all cases, the independent variable should be clearly identified – presumably the "known" quantities of graphite in the mixture (though this should be verified using total C analyses). The dependent variable should also be clearly identified. Lastly, while the calibration/regression lines are shown with error envelopes, none of the data points have errors associated with them. Were the analyses replicated at all? I found on mention of these. Clearly, each of the analytical methods has instrument/analytical error associated with them. How were these accounted for in the study?

4. The methods chosen for comparison were not especially the best available or most appropriate given the current literature. It almost seems as though there was a foregone conclusion that "smart combustion" was going to be the best and the need was to validate these. However, as the authors correctly point out, the use of "smart combustion" doesn't solve the problem of selecting a threshold temperature for distinguishing forms of organic C because it either uses the "wrong" temperature or at a minimum uses the same threshold for all samples. The selection of FTIR was intriguing since the presence of interference bands is well documented. I thought there might be a better spectral method (NIR, MIR, Raman, etc.) that would be better suited to the task. Similarly, the use of TGA has well documented shortcomings in that mass loss reactions are not all attributable to organic C combustion/pyrolysis. In fact, the only method that directly quantified carbon in the current study was the "smart combustion". While the authors discuss the possibility of combining methods, they seem to have missed the opportunity for using EGA during ramped heating - which is essentially what "smart combustion" is.

---

## Author Comment (AC1) · 5 Sep 2019

*I have read with interest the draft untitled "Identifying and quantifying geogenic organic carbon in soils – the case of graphte". Overall, I have found that the draft is very clear. To be published in SOIL, I consider that the authors should provide a proper description of the soils they used in the study. I have also several (rather minor) concerns and questions that have to be answered before I can recommend the publication of this draft in SOIL.*

Dear Reviewer,

First of all, we want to thank you for your time reviewing our article, your kind comments and your effort helping us improving the work. We will address your concerns and answer your questions, point by point.

- *I consider that the authors should provide a proper description of the soils they used in the study*

Thank you for addressing this important point. We indeed forgot the inclusion of a proper soil description. Hereby the revised part of the materials and method section:

"Top soil and fresh rock samples from a nearby outcrop, were taken from a field site in Rambla Honda, Sierra de los Filabres (37°07'43" N, 2°22'30" W / Southern Spain). The area is located in the Nevado-Filabride complex and contains Devonian-Carboniferous slaty micaschist with graphite and garnets crossed by abundant quartz veins (Puigdefábregas et al., 1996). Carbonates found in the soil sample (0.18 % C) originated from pedogenesis and dust deposition as the parent rock does not contain carbonates. Soil material was taken from the topsoil (0-5 cm, without sieving crust) under the grass tussock *Macrochloa tenacissima* to ensure a substantial amount of OC was present. The soil itself could be classified as Skeletic Leptosol (loamic colluvic) following the World reference base for soil resources (2014). Additional soil material was collected from a field near the town of Alboloduy (37°04'09" N, 2°36'43" W), hereafter referred to as AB soil, with similar vegetation and (climatic) conditions. The lithology consists of feldspathic mica schist (IGME, 1979), but without natural graphite and with a relative high CaCO3 content (1.87 % C). The AB soil was also sampled from the topsoil, without sieving crust, under the grass tussock *Macrochloa tenacissima*. According the World reference base for soil resources (2014) the soil could be classified as Skeletic Leptosol (loamic). The soil samples were dried at 40°C and sieved to ≤ 2 mm." Original p. 4, line 27 – p. 5, line 2

1) *Graphitic C can be found in rocks. Depending on P and T conditions experienced by the sediments, we do not get necessarily pure graphite. I am therefore wondering if the graphite standard material (Merck) is similar to the graphite found in the soils developed in micaschists. I would appreciate to see Raman signatures of the graphites used in the study. We can also imagine that some graphitic C with lots of defaults may evolved before the final oxidation step. In this case, such graphitic C would not be recovered in the ROC fraction. Can the authors discuss or rule out this hypothesis?*

It is right that pressure and temperature conditions are important factors determining the degree of graphitization. From literature we have an estimation what the conditions have been for the sampled rock, but the degree of graphitization of the standard is not provided by the manufacturer. We arranged Raman spectra (see Figure 1), which we will include in the revised version to shed light on potential differences between the standard and natural graphite as found in the rock and soil sample. As both the graphitic schist and the standard show a highly similar pattern, it can be assumed that they have a similar state of graphitization. The D1 (~1350 cm$^{-1}$), G (~1580 cm$^{-1}$) and D' (~1620 cm$^{-1}$) peaks indicated in Figure 1, can all be attributed to graphitic C, whereby the ratio between the D1 and the sum of all three peaks is a clear indication for the degree of graphitization (Beysacc et al., 2003; Ferrari, 2007). For the Standard (Merck) the ratio is 0.20, while for the graphitic schist we

obtained 0.34, which are both indicating well organized carbon (< 0.5, Beysacc et al., 2003). No peaks are observed around the 1200 and 1500 cm$^{-1}$ bands, including the soil sample, which would have indicated the presence of pyrogenic / black carbon components (Sadezky et al., 2005, Schmidt et al. 2002). The G-peak for the soil sample (calibration 1) coincides with the D' peak, as could be expected in a sample with different carbonaceous substances, but can also indicate defects in the crystalline graphite structure formed by weathering.

We will include the next paragraph to the method section together with the Raman spectra as in Figure 1:

"Raman spectra were made of the soil of calibration set 1, the standard (Merck) and graphitic schist, using a Thermo Scientific DXR Smart Raman Spektrometer, with 532 nm laser and a power output of 9 mW. Before the measurement, sample was pressed in aluminium cups. Peaks obtained were integrated using Lorentzian profiles fitting in Origin 2019.

In Figure 1 it can be seen that the spectra of the graphite standard (black) was highly similar to the graphitic schist (red). The D1 (~1350 cm$^{-1}$), G (~1580 cm$^{-1}$) and D' (~1620 cm$^{-1}$) peaks indicated in Figure 1, can all be attributed to graphitic C, whereby the ratio between the D1 and the sum of all three peaks is a clear indication for the degree of graphitization (Beysacc et al., 2003; Ferrari, 2007). For the graphite standard (Merck) the ratio is 0.20, while for the graphitic schist we obtained 0.34, which are both indicating well organized carbon (< 0.5, Beysacc et al., 2003). No peaks were observed around the 1200 and 1500 cm$^{-1}$ bands, including the soil sample (blue, Fig. 1), which would have indicated the presence of pyrogenic / black carbon components (Sadezky et al., 2005; Schmidt et al., 2002)"

[Figure]

**Figure 1. Raman spectra of the graphite standard (Black), graphitic schist (red) and soil of calibration set 1 (i.e. natural graphite containing soil, Blue). Vertical lines indicate the peaks for amorphous carbon (1342/1339 cm$^{-1}$) and peaks for graphitic carbon (1575 cm$^{-1}$ standard/schist and 1596 cm$^{-1}$ for soil of calibration set 1). Indicated are the D1 band (1350 cm$^{-1}$), caused by plane defects and heteroatoms in the carbon structure, G (1580 cm$^{-1}$), crystalline carbon i.e. pure graphite, and D' band (1620 cm$^{-1}$), caused by disordered graphitic lattices.**

Some graphitic C, especially with lots of defects or impurities in its mineral structure, might indeed evolve before the final oxidation step of the smart combustion method, resulting in an underestimation of the graphitic C content of the soil as it is not taken into the ROC fraction. We also hypothesized in the discussion that radicals, released from other minerals by temperatures of 700°C and higher, might induce graphite evolution under anoxic conditions (page 11, Line 28-33). According to the measurements with the smart combustion method about 6% of the total C was lost in samples of quartz + graphite standard (i.e. Calibration set 2), while the Graphitic schist lost 2% of the total C, although the graphite standard had a higher structural organization (lower ratio, as discussed above). We will extend line 28-33 (page 11) of the discussion to increase clarity about the loss of graphite before the final oxidation phase and include a Figure (Figure 8c below) visualizing this loss as follow:

"As shown by Hayhurst and Parmar (1998), very small impurities in the graphite can cause a small part of the graphite to pyrolyse during anoxic conditions at higher temperatures. Graphitic C of lesser graphitization, might therefore result in a larger loss of graphitic C during pyrolysis and a greater underestimation of the graphitic C content. Taking a closer look at the measurements of the artificial soil, reveals that a small part of the graphite started to oxidize under anoxic conditions (Fig. 8). The measurement of graphite in quartz, as in calibration set 2, showed that about 6% of the total carbon was lost during the pyrolysis phase, while for the graphitic schist this loss was 2% (Fig. 8c), resulting in an underestimation of the graphitic C content. Bews et al. (2001) suggested that at temperatures higher than 700 °C, radicals like $HO_2$ and OH might act as reactant with the pure C. Furthermore, in the method comparison study for recovering different types of black C, Roth et al. (2012) suggested a (relatively) strong catalytic effect of oxides on black C oxidation, which was most predominant in soils." p.11 line 28-33

[Figure]

**Figure 8. Examples of smart combustion measurements of the artificial soils (a), the fresh OM component (b), the graphite standard and graphitic schist (c).** The blue area delineates the part where $O_2$ is substituted for $N_2$ and the temperature program is displayed by the red dashed line. Note that artificial soil 2 (green) is without $CaCO_3$ and artifical soil 3 (orange) is without graphite.

2)  *2◦ Pyrogenic C (pyOC) can also resist to high T under anoxic conditions. In this case, some pyOC may be recovered in the ROC fraction. What would happen if the studied soil contains both graphite and pyOC? It may have been interesting to add charcoalin the tested mixtures. If the presence of pyOC is a limit to the method, it should be discussed.*

We understand from your question that we did not discuss sufficiently the potential interference of thermally resistant OM (like pyrogenic C / black C). Therefore, we will elaborate more on this topic in the discussion section, as showed below. Including other forms of thermally resistant C fractions and examine how we could distinguish between them would be a very interesting topic for further investigation. Nonetheless we can expect, with the current settings, that pyrogenic C might end up in the ROC fraction of the smart combustion method.

"When the sample contain other forms of thermally resistant OM or even black carbon, which are not pyrolyzed during the anoxic phase, this C component is likely to end up in the graphitic C fraction with the smart combustion method. Especially as most temperature boundaries are empirically derived (Pallasser et al., 2013; Ussiri et al., 2014), a pre-test with continues heating under oxic conditions, is therefore recommended to get an idea which/how many substances are present in the sample. According the Raman spectra (Fig. 1), no indications were found for the presence of black C in the soil and rock samples, as it should have created peaks / increased Raman intensity around the 1200 and 1500 cm$^{-1}$ bands (Sadezky et al., 2005, Schmidt et al. 2002). Further studies should focus on temperature boundaries of different substances in relation to their properties and see how for instance graphitic C can be distinguished from other thermally stable C components." After original p.12, line 11

3)  *10 samples for calibrating a model is definitely a too low number of samples. It would have been highly surprising to get nice results with such a low number of samples. It may have been interesting to use all the samples to design calibration models. We can't exclude that with a nice sample set containing 500 samples with known graphite concentrations, a convincing IR-based model can be designed.*

We agree that more samples could potentially improve the model. On the other hand, it is frequently shown in the literature that the performance of IR spectroscopic models for predicting soil properties increases with sample set homogeneity (e.g., Grinand et al., 2012), i.e., calibration and validation become more precisely when focussing on samples from similar or identical sites and soil matrixes. Here, we like to point out that two representative matrix substances were used as calibration samples: quartz sand and the soil of interest (soil 1), which was later used for validation. The $R^2$ and RMSEP of the calibrations were quite sufficient ($R^2$=0.96 and 0.99; RMSEP=0.24 and 0.10). The samples with the unknown graphite concentrations were of exactly the same matrix (Quartz, soil 1). So, the models we used were very specific in addition to the high $R^2$ and low RMSEP. Since further samples are not available, we calculated a model including both calibration data sets, soil 1 and quartz. This PLSR model used 3 components and an $R^2$ of 0.96 and an RMSEP of 0.24 (Fig. 2). These values were at the same level as found for the single models (see above). Nevertheless, all models substantially overestimated the graphite content. Therefore, we do not think that the use of more samples of different origin would improve the prediction / validation. Against the backdrop of the literature (specific graphite absorption bands that have been reported in the literature are only valid for oxidized graphite), the failed predictions of the graphite contents were plausible. We modified the discussion correspondingly, however, - if you agree - we do not intend to add the Figure (here shown as Fig. 2) to the text:

"The calibration between infrared spectra and graphite contents of the calibration sets yielded promising results (Figs. 1a and 1b) and could also be used for a cross-validation (Fig. 5). Although the same substrate materials and similar contents of graphitic C were used in the validation, the

graphite contents were systematically over-predicted. Despite the apparent quality of the calibration, this failure could have been caused by the relatively low number of calibration samples. Note that the use of the two calibration data sets, soil and quartz, in a joint PLSR model ($R^2 = 0.96$ and an RMSEP = 0.24; 3 components) did not improve the calibration nor the prediction accuracy. It cannot be excluded that a higher number of samples for the calibration could improve the PLSR model and the prediction results. Further, Raman spectroscopy might be an alternative approach for quantifying graphite in soil samples (e.g., Sparkes et al., 2013; Jorio and Filho, 2016)." Original line p.9, 17-20

[Figure]

**Figure 2. Prediction plot of the PLSR model using a joint dataset of soil 1 and quartz.**

4) *The authors hypothesized that ROC content would match graphite content. It is not too far but not perfect. Why don't the authors try to design a model based on ROC results as they did with IR and TGA results?*

Thank you for the suggestion to create a model for correcting the ROC value. We considered this as well, but, as can be seen in supplementary Figure 4 of the manuscript, the total carbon measured with the smart combustion method (i.e. the Soli-TOC device) is the same as what is obtained by the more accurate elemental analyser. Therefore, the slight underestimated graphitic C content results from the differentiation between carbon fractions by the temperature-oxidation program and not as a results of the direct output of the sensor. Furthermore, correcting the ROC/graphitic C value using the calibration set would not improve the graphitic C estimation as a model build with calibration set 1 would result in a slight overestimation of calibration set 2 (notice slope > 1.00) and vice versa (see Figures 3 and 4, here below). This difference in underestimation of the ROC fraction was attributed to impurities in the graphite and/or presence of radicals, as discussed in the second paragraph of section 4.3. Especially the presence in radicals will differ from sample to sample, as it depends on the matrix composition, which would give the same matrix issue in the model creation

as with the FTIR and TGA methods. Therefore, we suggest keeping the ROC content as it is derived by the device and clarify this decision in the method section by adding:

"The Soli-TOC device directly converts the NDIR signal in the C content of the different components, as calibrated with CaCO3. Creating an additional model to correct the C output, is introducing an additional error in the measurements. Therefore, we analysed the direct C output, as measured in the ROC fraction. Triplicate measurements were averaged, whereby the average coefficient of variation between replicates was 2.7%, and a Pearson correlation test was performed between the obtained ROC data and calibration sets to analyse how well the graphite content was measured." Original p.7, line 5-7

[Figure]

**Figure 3. Correction of the ROC values by using a model based on calibration set 1 (graphitic soil with added graphite). In orange the original measured graphitic C content (ROC) of calibration set 1 is plotted against the added graphite. As can be seen by the linear trend line, the graphitic C content is originally overestimated. A simple correction model was created, resulting in an exact estimation (Blue), but the same model resulted in an underestimation of graphitic C in calibration set 2 (Grey, Quartz with added graphite).**

[Figure]

**Figure 4. Correction of the ROC values by using a model based on calibration set 2 (Quartz with added graphite). In yellow the original measured graphitic C content (ROC) of calibration set 2 is plotted against the added graphite. As can be seen by the linear trend line, the graphitic C content is originally slightly underestimated. A simple correction model was created, resulting in an exact estimation (purple), but the same model resulted in an overestimation of graphitic C in calibration set 1 (Green, Quartz with added graphite).**

5) *I do not understand the Figure 5. I suggest improving the explanations on this Figure or removing it*

We hope that Figure 5 in the manuscript, which summarizes the results of the three tested methods, becomes clearer with this extended Figure description:

[Figure]

"Figure 5: Overview of the predicted amount of graphite in the calibration sets (squares/diamonds), artificial soil (inset, circles/triangle), graphitic schist (inset, stars) and AB soil (right pointing triangle) as measured with the different methods. Black symbols: graphite prediction by FTIR, model from calibration set 1; White: FTIR, calibration set 2; Orange: graphite prediction by TGA, model from calibration set 1; Green: graphite prediction by TGA, model from calibration set 2; Grey: graphite prediction by smart combustion. Exact data is given in Table 2."

References

Beyssac, O., Goffé, B., Petitet, J. P., Froigneux, E., Moreau, M., Rouzaud, J. N. 2003. On the characterization of disordered and heterogeneous carbonaceous materials by Raman spectroscopy. Spectrochimica Acta Part A: Molecular and Biomolecular Spectroscopy, 59(10), 2267-2276.

Ferrari, A.C. 2007. Raman spectroscopy of graphene and graphite: Disorder, electron–phononcoupling, doping and nonadiabatic effects. Solid State Communications 143, 47–57.

Grinand, C. ,Barthès, B.G., Brunet, D. , Kouakoua, E. , Arrouays, D., Jolivet, C., Cariac, G., Bernoux, M. 2012. Prediction of soil organic and inorganic carbon contentsat a national scale (France) using mid-infrared reflectancespectroscopy (MIRS). Eur. J. Soil Sci. 63, 141 - 151.

Jorio, A. and A.G.S. Filho 2016. Raman Studies of Carbon Nanostructures. Annual Review of Materials Research 46, 357-382.

Sadezky, A., Muckenhuber, H., Grothe, H., Niessner, R., Pöschl, U. 2005. Raman microspectroscopy of soot and related carbonaceous materials: spectral analysis and structural information. Carbon, 43(8), 1731-1742.

Schmidt, M. W., Skjemstad, J. O., Jäger, C. 2002. Carbon isotope geochemistry and nanomorphology of soil black carbon: Black chernozemic soils in central Europe originate from ancient biomass burning. Global Biogeochemical Cycles, 16(4), 70-1.

Sparkes, R., Hovius, N., Galy, A., Kumar, R.V., Liu, J.T. 2013. Automated analysis of carbon in powdered geological and environmental samples by Raman Spectroscopy. Applied Spectroscopy 67, 779-788.

World Reference Base for Soil Resources 2014. IUSS Working Group WRB. International soil classification system for naming soils and creating legends for soil maps. World Soil Resources Reports No. 106. FAO, Rome.

---

## Author Comment (AC2) · 5 Sep 2019

*This is a very well written manuscript that reports a methods comparison for the identification/ quantification of graphite in soil. The writing is excellent, and the reporting of results is clear. But the study have a few important shortcomings that I'd like to see addressed before I would deem it acceptable for publication.*

Dear Reviewer,

First of all, we want to thank you for your time reviewing our article and your effort helping us improving the work. Following the numbering in your review, we provide answers to your concerns and questions.

1. *The manuscript does not provide sufficient context for focusing exclusively on graphite. The reader might interpret the current rationale as a narrow justification for using graphite in the experiment. I understand that graphite may form from metamorphic processes, but is the goal of the study to quantify graphite specifically, or geogenic organic C more broadly? The former seems far too narrow a prospect given the wide range of forms of geogenic C, and the latter is underdeveloped in the study. Without further elaboration, graphite seems a little too specific. Galy, Hemingway and others are not specific when they refer to "lithospheric" C, Ussiri (cited in the manuscript), Chan (2017, Themochimica Acta) and others have targeted coal, and there is a large and growing literature on pyrogenic C in soils. How would the presence of these affect results? Can graphite be distinguished from other forms of thermally recalcitrant organic C? The authors do well to distinguish between carbonates (which varies widely among dolomite, calcite, etc.) and thermally recalcitrant C, but have not adequately elaborated on graphite versus other forms of geogenic C, let alone pyrogenic C. The distinction between the latter two is obvious using 14C, but the issue here is among geogenic C forms.*

Yes, you're right that we want to develop a quantification method just for graphite. As discussed by for instance Ussiri et al. (2014), there is a wide continuum of geologically altered organic compounds, whereby a general quantification method seems not to be possible as spectral and thermal properties gradually change by the degree of transformation of the organic matter. Furthermore, Roth et al. (2012) showed for several black carbon types that there is no ideal method to quantify all their tested black carbon types, especially in soil environments. As to our knowledge no previous study has attempted to quantify graphitic carbon, especially not in a soil environment, it hampers studying carbon dynamics in soils developed on sites with graphite containing parent materials, as experienced by ourselves. Therefore, we will re-write parts of the introduction to put more the focus on graphitic carbon:

"Organic C (OC) of geogenic origin, which gained less attention until now, is formed when organic compounds in sediments undergo coalification or kerogen transformation during diagenesis. Under high pressure and appropriate temperature conditions this process can continue into the formation of graphitic C, although well-crystallized pure C is hardly reached (Oohashi et al., 2012; Buseck and Beysacc, 2014). Redox transformations during metamorphoses of carbonates leads also to the formation of highly crystalline graphite (Galvez et al., 2013). Intruding hydrothermal fluids in the earth's crust forms a third source of graphitic C during rock formation, which produces the purest graphite crystals (Rumble, 2014). This relative pure and stable form of C is highly chemical inert, although impurities from the parent material increases its chemical reactivity (Beyssac and Rumble, 2014). Via tectonic processes graphite bearing rocks can reach the earth surface where they are subjected to physical and chemical weathering. Therefore, graphitic C occurs mainly in rocks from orogenic belts and in metasedimentary rocks in old cratons and might be a quite common bedrock for soil development (Hartmann and Moosdorf, 2012; Buseck and Beysacc, 2014).

The fate of geogenic graphite under weathering and soil formation has rarely been studied, possibly due to the lack of methods for determining and quantifying geogenic graphite beyond the background of soil OC (OC). There are some indications that a substantial part of the geogenic graphitic

C is actually lost in the pathway from rock weathering to (marine) sedimentation (Galy et al., 2008; Clark et al., 2017). Isolated naphthalene-degrading bacteria from contaminated soil proofed to oxidize and degrade graphitic materials, questioning the assumed biological inactivity of graphite (Liu et al., 2015). In a recent study, Hemmingway et al. (2018) estimated 2/3 of the graphitic C to get oxidized during soil formation, strongly facilitated by soil microbial activity." p.2, line 12-23

To answer your question if graphite can be distinguished from other forms of thermally recalcitrant organic C, we can be sure that with smart combustion method, in its current settings, is certainly not capable to do so as it lumps all the oxidizable carbon components that evolve between 400 and 900°C in one fraction. From FTIR spectroscopy, a valid proof of the existence of pyrogenic C in the soil samples is not possible. For a clear evidence, spectrometric techniques such as e.g. Pyrolysis-Field Ionization Mass Spectrometry (Py-FIMS) would be necessary, as shown in Leue et al. (2016), which were beyond the scope of the study.

Furthermore, we arranged Raman spectra of the soil of calibration set 1 and graphitic schist samples, as shown in Figure 1 at the end of this letter. The D1 (~1350 cm-1), G (~1580 cm-1) and D' (~1620 cm-1) peaks indicated in Figure 1, can all be attributed to graphitic C, whereby the ratio between the D1 and the sum of all three peaks is a clear indication for the degree of graphitization (Beysacc et al., 2003; Ferrari, 2007). The Raman spectra showed no signs of pyrogenic or black carbon in both the soil sample (Fig. 1), which would have created a peak at 1200 cm$^{-1}$ and/or a clear shoulder at 1500 cm$^{-1}$ (Sadezky, 2005; Schmidt et al., 2002). Nonetheless it is a good point that care should be taken to distinguish between pyrogenic/black carbon and the graphitic C. Therefore, we will include in the discussion section more clarification on this point:

"When the sample contain other forms of thermally resistant OM or even black carbon, which are not pyrolyzed during the anoxic phase, this C component is likely to end up in the graphitic C fraction with the smart combustion method. Especially as most temperature boundaries are empirically derived (Pallasser et al., 2013; Ussiri et al., 2014), a pre-test with continues heating under oxic conditions, is therefore recommended to get an idea which/how many substances are present in the sample. According the Raman spectra (Fig. 1), no indications were found for the presence of black C in the soil and rock samples, as it should have created peaks / increased Raman intensity around the 1200 and 1500 cm$^{-1}$ bands (Sadezky et al., 2005, Schmidt et al. 2002). Further studies should focus on temperature boundaries of different substances in relation to their properties and see how for instance graphitic C can be distinguished from other thermally stable C components." After p.12, line 11

Concerning the mentioned literature, we want to clarify our choice for them hereby.

- Galy et al. (2008), doi: 10.1126/science.1161408, uses different sediment (samples) to study the origin of petrologic carbon. They state that part of the graphite, present in the parent material, is no longer present in the downstream sediments based on Raman spectra and transmission electron microscopic images, indicating that it has been oxidized during the erosion/weathering process. Although the degree of graphitization is important for the preservation, as mainly the lesser graphitized carbon was lost, we considered this reference as a good case study to show that graphite is somewhere lost (i.e. used) in the weathering process of the parent material, while this has hardly been documented or studied.
- Hemingway et al. (2018), doi: 10.1126/science.aao6463, although the title of their paper indicate only lithographic C, a more detailed description of the lithology of the study area revealed that they were dealing with graphite containing metamorphic rocks, whereby the rock contained graphite of different degrees of graphitization. More detailed information can be found in the supplementary of the Hemingway article and in the paper of Hilton et al. (2010), doi.org/10.1016/j.gca.2010.03.004

- Ussiri et al. (2014), doi: 10.1016/j.geoderma.2013.09.015, have made a comprehensive review on the current available methods and definitions to distinguish geogenic carbon (mainly coal) from other carbon sources, including inorganic carbon. They also consider that "coalification" (i.e. early stage of graphitization) is a process, whereby the stages of coalification determine their susceptibility for a certain analytic method. They furthermore emphasize that no standardized method exists to identify and separate geogenic carbon from other carbon sources in soils. In this context we used this paper, as, to our knowledge, no other comprehensive discussion on distinguishing of (geogenic) organic and inorganic carbon in soil samples exist.

Thank you for suggesting the paper of Chan et al. (2017), doi: 10.1016/j.tca.2017.02.006, as we were not aware of this study. However, the software they applied to process their TGA measurements further into the endmembers is not available to us and could therefore not be tested.

*2. As a result of the discussion above, and more generally for a method development study, I found the number and range of materials used too small. Only one graphite-containing natural soil and one carbonate-containing natural soil were used for validation. The artificial soil mixtures were made from one OM source far removed from the natural soil (Germany vs. Spain), and "neat" mineral specimens (quartz, muscovite, CaCO3 (not dolomite or calcite?)). While I understand the desire to create a reductionist, simplified system for initial testing, the result is only nine samples of one mixed-matrix to generate the calibration. This is a highly undersampled relationship. This is critically important because the authors are right to highlight matrix effects, but they do not adequately account for these in the design of the calibration/correlation study. CaCO3 is not dolomite or calcite, soil minerals are often interstratified, and graphite likely exists in a mineral-associated form fused to the mineral matrix. None of these incipient properties of soil are accounted for in the method development – making the results of limited value. It is a nice proof of principle, but the study needs to go well beyond this given the current state of the literature. I don't see how this study substantially move us further than some of the studies cited within it.*

As we are focussing on developing a quantification method for graphite in soil matrix, we started with creating samples of pure quartz mixed with different quantities of graphite to test the methods available to us. The next step was proofing that we could identify/distinguish graphite from other typical present substances in the soil, for which we created the artificial samples. To our opinion OM from a forest floor, although geographically not close to the natural samples, is still providing a typical input signal for OM as would be found in rangeland soils from Southern Spain.

Another important carbon component in semiarid soils is carbonate, mostly in the form of calcite. It is true that the used pure $CaCO_3$ is not exactly same as pedogenic or geogenic calcite, but for the simplified artificial soil and to test the differentiation between carbonates and graphitic C it should be sufficiently similar. The most important difference should be visible in the thermal properties, whereby pedogenic carbonates tend to start decomposing at 550-600°C, with the major decomposition peak coming around 750°C (e.g. Apesteguia et al. 2019; Pallaser et al. 2013), while the purer calcite started with decomposition just above 600°C, it reached its major decomposition peak around 725°C (Fig. 2 in the manuscript). Note that we also include an additional soil, "AB Soil", which contains a large amount of pedogenic calcite. In the Figure 5 of the manuscript it can be seen that there is no difference in predicted graphite between the AB soil and the quartz, both spiked with graphite standard. This indicates that there is no significant influence of the pedogenic carbonates on the graphite prediction with the smart combustion method.

In the next step we tested natural soil and graphitic rock. By creating a sample set with different amounts of graphite added to the soil, we tested the methods for their ability to quantify graphite. By taking also a carbonate rich soil with a different mineral composition (i.e. feldspathic and without garnets) we also took the influence of mineralogy on the ability of graphite quantification into account, which resulted in the matrix effect highlighted in the manuscript.

In our point of view, further study is only realistic using smart combustion or a comparable method (like EGA or Rock-Eval) as they proofed to be most promising. The alternating between oxic and anoxic conditions during a measurement is also a not often employed method to differentiate between soil carbon components. For FTIR it is frequently shown in the literature that the performance

of IR spectroscopic models for predicting soil properties increases with sample set homogeneity (e.g., Grinand et al., 2012), i.e., calibration and validation become more precisely when focussing on samples from similar or identical sites and soil matrixes. Calculating a model including both calibration data sets, soil 1 and quartz resulted in a PLSR model with an $R^2$ of 0.96 and an RMSEP of 0.24. These values were the same level as found for the single models. Nevertheless, all models substantially overestimated the graphite content. We will highlight this even further in the discussion (see below and 2nd addition under point 1). The use of graphite addition might be most practical for testing quantification of graphitic C in different mineral matrixes. This should also shed more light on how geogenic C, pyrogenic C and carbonates could be distinguished from each other.

"The calibration between infrared spectra and graphite contents of the calibration sets yielded promising results (Figs. 1a and 1b) and could also be used for a cross-validation (Fig. 5). Although the same substrate materials and similar contents of graphitic C were used in the validation, the graphite contents were systematically over-predicted. Despite the apparent quality of the calibration, this failure could have been caused by the relatively low number of calibration samples. Note that the use of the two calibration data sets, soil and quartz, in a joint PLSR model ($R^2$ = 0.96 and an RMSEP = 0.24; 3 components) did not improve the calibration nor the prediction accuracy. It cannot be excluded that a higher number of samples for the calibration could improve the PLSR model and the prediction results. Further, Raman spectroscopy might be an alternative approach for quantifying graphite in soil samples (e.g., Sparkes et al., 2013; Jorio and Filho, 2016)." Original p.9, line 17-20

3. *The authors identified one of the challenges of thermally distinguishing forms of C as the determination of threshold temperatures. However, the description of how TGA data was processed is inadequate. Phrases like "models were created from the calibrations sets" and "the best temperature limits for quantification of graphitic C in the calibration sets was determined" are not reproducible. These steps may be the most critical step of the process, but even the most experienced expert in this field would be unable to verify and repeat it. More detail is required here. There are also no details provided on how these data were used in the calibration. The selection of different threshold temperatures somewhat undermines the "smart combustion" approach if it cannot be universally applied. Perhaps there is some elaboration required in the discussion about how much control over threshold temperature there is available with such an instrument, and whether the DIN methods are suitable/adequate standards.*

We are sorry that the data processing was not clearly stated. Together with your next point, we have extended and revised our description of the data processing, which is hopefully now better reproducible (see point 4 for suggested revision).
    We will emphasize our discussion of the DIN method to highlight that the smart combustion method is a rather standardized version of the EGA method and that it should therefore be applied with care as long as the thermal boundaries between substances are unknown.

"Although we focused in this study on the ROC component, which significantly correlated with the graphite content, considering the other components in the DIN19539-standard was beyond the scope of this study. Nonetheless, we found indications that the thermal boundaries defined in the DIN19539-standard are not ideal to differentiate between soil OM and inorganic C (Fig. 8). As most carbonates start to decompose at temperatures of 550°C (Földvári, 2011), it might be more suitable to increase the level for the TOC component from 400 to 500°C. Only when black C is present in the sample, which might oxidize between 375 and 540 C (Roth et al., 2012), this might lead to an overestimation of the TOC content. Using TGA simultaneously with differential scanning calorimetry, water and $CO_2/H_2O$ flux measurements (i.e. evolved gas analysis, Fernández et al., 2012) or with the Rock-Eval method focusing on hydrocarbon, $CO_2$ and CO release (Behar et al., 2001), could improve the development for

a more standardized method applicable to soils using combustion elemental analysers. The overlap between the thermal properties of different C components emphasizes the need to always first consider what is present in the sample and what might interfere with the considered applied methodology, before applying a fast and standardized analytic method." p.12, line 25

4. *I would argue that since this is such a key/core component of the study, that it deserves its own separate subsection within the methods (ie, statistical analyses). I'm not 100% sure that my comments here will be relevant or correct because it was difficult to follow precisely how each of the calibrations were generated. But if I read it correctly, the authors appear to use different methods to generate the calibration curves depending on the quantification method. PLS was used for the FTIR method of quantification, not sure how it was done for what data from TGA, and Pearson correlation for the "smart combustion" data. In all cases, the independent variable should be clearly identified – presumably the "known" quantities of graphite in the mixture (though this should be verified using total C analyses). The dependent variable should also be clearly identified. Lastly, while the calibration/regression lines are shown with error envelopes, none of the data points have errors associated with them. Were the analyses replicated at all? I found on mention of these. Clearly, each of the analytical methods has instrument/analytical error associated with them. How were these accounted for in the study?*

As every method requires its own statistical analysis, we decided to split the statistical analysis section over the different method sections. As this seems not to make it clearer, we will create subsections for each method, clearly stating the statistical analysis conducted. The instrumental error for the TGA analyser has to be checked with the technician, who is at the moment of writing unavailable, but will be added in the revised manuscript. Furthermore, we have extended the part of the TGA method to increase clarity on the data processing:

"2.2.1 Statistical analysis of the FTIR spectroscopic data
The partial least squares regression (PLSR) analyses of correlations between the transmission or DRIFT spectra and the graphite contents (0.1 - 4 %) of the samples were performed using R, Version 3.1.1 (R Core Team, 2014) with module PLS (SIMPLS, cross-validation: leave-one-out) of Mevik et al. (2018). The signal intensities were used as independent variables, the graphite content as dependent. The number of components used in the calibration models followed the lowest predicted root-mean-square error (RMSEP) of the specific datasets. The scores and loadings were plotted for the two main components determining most of the variances of the DRIFT spectra. Larger absolute loading values of signal intensities in certain WN regions imply a greater importance of these WN for the cumulated values of the principal component 1 or 2 displayed in the score plot." Original line 31 page 5 – line 5 page 6

"2.3.1 Statistical analysis of the TGA data
TGA measurements were processed and thermal mass loss data obtained via the Proteus Thermal Analysis software (NETZSCH, Hanau, Germany). Measured sample mass data are fitted with the spline function over the temperature, with steps of 1 °C. Further analyses of the obtained data were conducted using R, Version 3.5.1 (R Core Team, 2018). Using the module PLS (cross-validation: leave-one-out) of Mevik et al. (2018), a model was created to determine the graphite content based on the measured mass change in a certain temperature range for each calibration set and the known graphite addition, using the mass change as independent variable. By iterating the model creation over the temperature range from 400 to 1075 °C, with minimum step of 5°C difference, and recording the slope, intercept and RMSEP of each created model, it was explored which temperature range described best the graphite content of both the calibration sets. The RMSEP of these models were checked, which is visualized in supplementary 3, and a single temperature range that fit both calibration sets, was determined." Original line 17-21 page 6.

"2.4.1 Statistical analysis of the Smart Combustion data
The Soli-TOC device directly converts the NDIR signal in the C content of the different components, as calibrated with $CaCO_3$. Creating an additional model to correct the C output, is introducing an additional error in the measurements. Therefore, we analysed the direct C output, as measured in the

ROC fraction. Triplicate measurements were averaged, whereby the average coefficient of variation between replicates was 2.7%, and a Pearson correlation test was performed between the obtained ROC data and calibration sets to analyse how well the graphite content was measured." Original p.7, line 5-7

5. *The methods chosen for comparison were not especially the best available or most appropriate given the current literature. It almost seems as though there was a foregone conclusion that "smart combustion" was going to be the best and the need was to validate these. However, as the authors correctly point out, the use of "smart combustion" doesn't solve the problem of selecting a threshold temperature for distinguishing forms of organic C because it either uses the "wrong" temperature or at a minimum uses the same threshold for all samples. The selection of FTIR was intriguing since the presence of interference bands is well documented. I thought there might be a better spectral method (NIR, MIR, Raman, etc.) that would be better suited to the task. Similarly, the use of TGA has well documented shortcomings in that mass loss reactions are not all attributable to organic C combustion/pyrolysis. In fact, the only method that directly quantified carbon in the current study was the "smart combustion". While the authors discuss the possibility of combining methods, they seem to have missed the opportunity for using EGA during ramped heating - which is essentially what "smart combustion" is.*

We encountered an analytical problem during our work by the standard methods employed for carbon quantification and recognized the need for the development of a quantitative method to determine graphite in soils. Therefore, we developed a test program with methods available to us to overcome this issue. As we expect that others might encounter similar problems (because of the occurrence of graphite containing rocks as the base for soil formation), we decided to share our experience with the scientific community.

It is completely right that mass loss obtained by the TGA method are difficult to correlate with organic C, as OM tends to be chemically heterogeneous. Nonetheless, graphitic C is almost pure C and by using graphite addition tests, we expected the TGA method to correlate better with the graphitic C content than with OM content of soil samples.

Smart combustion, in our case with the Soli-TOC device of Elementar company, is indeed not the most flexible method to explore thermal properties of different carbonaceous substances as provided with most EGA methods. On the other hand, the smart combustion method provides a standardized set-up making the measurement of larger sample sets more feasible. We will highlight this further in the discussion section 4.4, as given under point 3.

Concerning the potential suitability of Raman spectroscopy, we added a sentence to the discussion. We want to notice that NIR and MIR (suggested by the reviewer as suitable for graphite determination) are FTIR techniques. As mentioned in the methods, we used FTIR spectroscopy in the mid-infrared wavelengths (2.5 – 25 μm). Unfortunately, Raman spectroscopy was not fully available for this study. Nonetheless we will add the Raman spectra below for clarification about the degree of graphitization of the graphitic schist / standard graphite and to show that there were no indications for the presence of pyrogenic C, as discussed under point 1. Furthermore, we will include a few sentences on the potential of Raman spectroscopy for developing a quantification method:

"As Raman spectroscopy is well able to distinguish graphitic C and determine its degree of graphitization, it seems to be a promising method. Nonetheless to use Raman spectrometry for quantification of substances in a soil matrix, further studies should first focus on standardization of sample preparation, as it has a large influence on the measured intensities and baseline determination and thereby the direct quantification of components (Beysacc and Lazzeri, 2012; Sparkes et al., 2013)." Between p.12, line 24/25

[Figure]

**Figure 1. Raman spectra of the graphite standard (Black), graphitic schist (red) and soil of calibration set 1 (i.e. natural graphite containing soil, Blue). Vertical lines indicate the peaks for amorphous carbon (1342/1339 cm⁻¹) and peaks for graphitic carbon (1575 cm⁻¹ standard/schist and 1596 cm⁻¹ for soil of calibration set 1). Indicated are the D1 band (1350 cm⁻¹), caused by plane defects and heteroatoms in the carbon structure, G (1580 cm⁻¹), crystalline carbon i.e. pure graphite, and D' band (1620 cm⁻¹), caused by disordered graphitic lattices.**

References

Apesteguia, M., Plante, A. F. and Virto, I. 2018: Methods assessment for organic and inorganic carbon quantification in calcareous soils of the Mediterranean region. Geoderma regional, 12, 39-48, doi: 10.1016/j.geodrs.2017.12.001.

Beyssac, O., Lazzeri, M. 2012. Application of Raman spectroscopy to the study of graphitic carbons in the Earth Sciences. Applications of Raman spectroscopy to earth sciences and cultural heritage. EMU Notes in Mineralogy, 12, 415-454.

Galvez, M. E., Beyssac, O., Martinez, I., Benzerara, K., Chaduteau, C., Malvoisin, B., and Malavieille, J. 2013: Graphite formation by carbonate reduction during subduction. Nature Geoscience, 6(6), 473, doi: 10.1038/NGEO1827.

Grinand, C. ,Barthès, B.G., Brunet, D. , Kouakoua, E. , Arrouays, D., Jolivet, C., Cariac, G., Bernoux, M. 2012. Prediction of soil organic and inorganic carbon contentsat a national scale (France) using mid-infrared reflectancespectroscopy (MIRS). Eur. J. Soil Sci. 63, 141 - 151.

Leue, M., K.-U. Eckhardt, R.H. Ellerbrock, H.H. Gerke, and Leinweber, P. 2016. Analyzing organic matter composition at intact biopore and crack surfaces by combining DRIFT spectroscopy and Pyrolysis-Field Ionization Mass Spectrometry. J. Plant Nutr. Soil Sci., 179:5–17.

Hartmann, J. and Moosdorf, N., 2012: The new global lithological map database GLiM: A representation of rock properties at the Earth surface. Geochemistry, Geophysics, Geosystems, 13(12), doi: 10.1029/2012GC004370.

Hilton, R. G., Galy, A., Hovius, N., Horng, M. J., Chen, H. 2010. The isotopic composition of particulate organic carbon in mountain rivers of Taiwan. Geochimica et Cosmochimica Acta, 74(11), 3164-3181.

Pallasser, R., Minasny, B. and McBratney, A. B. 2013: Soil carbon determination by thermogravimetrics. PeerJ, 1, e6, doi: 10.7717/peerj.6.

Roth, P. J., Lehndorff, E., Brodowski, S., Bornemann, L., Sanchez-García, L., Gustafsson, Ö. and Amelung, W. 2012: Differentiation of charcoal, soot and diagenetic carbon in soil: Method comparison and perspectives. Organic Geochemistry, 46, 66-75, doi: 10.1016/j.orggeochem.2012.01.012.

Sadezky, A., Muckenhuber, H., Grothe, H., Niessner, R., Pöschl, U. 2005. Raman microspectroscopy of soot and related carbonaceous materials: spectral analysis and structural information. Carbon, 43(8), 1731-1742.

Schmidt, M. W., Skjemstad, J. O., Jäger, C. 2002. Carbon isotope geochemistry and nanomorphology of soil black carbon: Black chernozemic soils in central Europe originate from ancient biomass burning. Global Biogeochemical Cycles, 16(4), 70-1.

Sparkes, R., Hovius, N., Galy, A., Kumar, R.V., Liu, J.T. 2013. Automated analysis of carbon in powdered geological and environmental samples by Raman Spectroscopy. Applied Spectroscopy 67, 779-788.

Ussiri, D. A., Jacinthe, P. A. and Lal, R. 2014: Methods for determination of coal carbon in reclaimed minesoils: a review. Geoderma, 214, 155-167, doi: 10.1016/j.geoderma.2013.09.015.